# CTTS: Collective Test-Time Scaling

## Abstract

Test-time scaling (TTS) has emerged as a promising, training-free approach for enhancing large language model (LLM) performance. However, the efficacy of existing methods, such as Best-of-N and Self-Consistency, is fundamentally constrained by the dominant single test-time scaling (STTS) paradigm, which relies on a single LLM agent interacting with a single reward model (SA-SR). Inspired by recent work showing that collective methods can surpass the performance ceiling of individual models, we introduce **Collective Test-Time Scaling (CTTS)**. First, we systematically investigate three primary interaction paradigms of existing multiple models: single-agent-multi-reward (SA-MR), multi-agent-single-reward (MA-SR), and multi-agent-multi-reward (MA-MR). Extensive experiments reveal that the MA-MR paradigm is consistently superior. Based on this finding, we further propose **CTTS-MM**, a novel framework that operationalizes multi-agent and multi-reward collaboration. CTTS-MM integrates two key technical contributions: (1) for agent collaboration, an **Agent Collaboration Search (ACS)** that identifies the most effective combination of LLMs from a candidate pool; and (2) for reward model collaboration, a **Mixture of Reward Models (MoR)** strategy that leverages a Prior Reward model Ensemble Selection (PRES) algorithm to select the optimal ensemble. Evaluations across seven mainstream benchmarks demonstrate that CTTS-MM significantly outperforms leading STTS methods (**+4.82% over Best-of-N**) and surpasses even flagship proprietary LLMs (**+7.06% over GPT-4.1**) and open-source LLMs. These results highlight the substantial potential of collective scaling to push the frontier of LLM inference.

## 1 Introduction

Recent advancements in large language models (LLMs) OpenAI (2025); Yang et al. (2024b); Brown et al. (2020); DeepSeek-AI & et al. (2025); Touvron et al. (2023) have marked a significant milestone in natural language understanding and generation. LLMs are typically optimized through training-time scaling, where huge amounts of data and parameters are applied, facing growing limitations due to their resource-intensive nature and the endless hunger for human data. To avoid introducing an extra expensive training process, test-time scaling (TTS) has emerged as an orthogonal direction for fully stimulating the ability of pre-trained LLMs during inference. The process of typical TTS methods Snell et al. (2025); Wang et al. (2023); Brown et al. (2024); Madaan et al. (2023), i.e., self-repetition-based methods Snell et al. (2025); Brown et al. (2024) can be divided into two sequential stages: 1) an LLM agent generates multiple candidate answers; 2) an external selector (reward model or manually designed selection metric) chooses the best answer. The performance of TTS highly relies on the inference quality of the LLM agent and the selection accuracy of the selector. Currently, TTS methods primarily adopt single TTS paradigm that consists of a single agent with a single selector (referred to as a "single to single" framework), which introduces two major limitations: 1) In the first stage, the limited capacity of a single agent causes a biased output distribution, leading to a constrained performance upper bound; 2) In the second stage, it imposes a prior selection preference, which hinders comprehensive and high-quality scoring of candidate answers. These intrinsic limitations of the "single to single" framework impede the further performance improvement of TTS and even lead to collapse. Thus, an essential question naturally arises: **How can TTS overcome the "single to single" framework to release the potential of LLM inference?**

Human behavior may offer some insights into the question. When tackling problems, people often engage in collaboration within teams to reach better solutions. Further, particularly challenging tasks

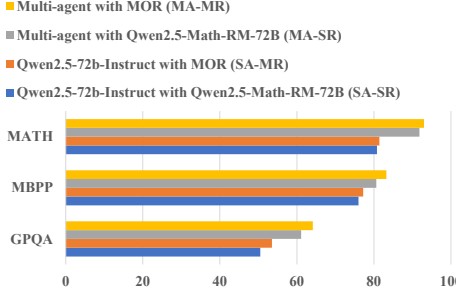

Figure 1: Comparison of previous Single Test-time Scaling (STTS) paradigm and our proposed novel Collective Test-time Scaling (CTTS) paradigms.

may require cooperation across multiple groups, combining diverse perspectives to aggregate a more comprehensive and effective outcome. This pattern is also reflected in recent developments of collective methods Wang et al. (2025); Chen et al. (2025); Shnitzer et al. (2024); Lu et al. (2024). For instance, Mixture-of-Agents (MoA) Wang et al. (2025) exploits the references from diverse LLM agents to aggregate a higher-quality final answer, breaking through the upper bound of single-agent systems. Inspired by collective methods, we advance the previous **Single TTS (STTS)** to the novel **Collective Test-Time Scaling (CTTS)**, taking the first step to explore the potential of CTTS. Specifically, we focus on investigating two key questions: (1) What is the optimal paradigm of collective test-time scaling? (2) How can we effectively scale the systems under

Figure 2: Comparison of three CTTS paradigm and one STTS paradigm on MATH, MBPP and GPQA.

such paradigm? To address the two questions above, we systematically design and explore three CTTS paradigms: (1) **single agent to multiple reward models (SA-MR)**; (2) **multiple agents to single reward model (MA-SR)**; (3) **multiple agents to multiple reward models (MA-MR)**. Figure 1 illustrates the differences between our proposed CTTS paradigms and the existing single TTS paradigm. To obtain the optimal CTTS paradigm, we conduct experiments to compare the above four paradigms under three benchmarks. The results are shown in Figure 2. It can be observed that, as the collective level increases, the performance improves, and the MA-MR paradigm consistently achieves the most substantial performance improvements, highlighting that both multi-agent and multi-reward-model collaboration play a critical role in the LLM inference performance.

Based on this observation, we adopt CTTS with MA-MR paradigm as the basic framework and propose a novel CTTS method called Collective Test-Time Scaling with Multiple agents to Multiple reward models (**CTTS-MM**) as an effective and simple specific instance of MA-MR paradigm. Specifically, for multi-agent collaboration specific to TTS, we first employ an **Agent Collaboration Search (ACS)** to choose the most effective agent ensemble from a candidate model pool. To guide the search with high-quality feedback, we propose a **Mixture of Reward Models (MoR)** to achieve multi-reward-model collaboration that breaks through the upper bound of a single reward model. To construct MoR regarding the given question, Prior Reward model Ensemble Selection (PRES) is proposed to select the optimal reward model or a weighted combination of them based on Pairwise Reward Ranking (PRR) metric over a curated question pool. To verify the effectiveness of our proposed CTTS-MM, we conduct extensive experiments on seven mainstream benchmarks with ten open-source LLM agents and eight reward models. Compared with existing popular TTS, collaboration methods and leading LLMs, CTTS-MM achieves significant superiority. For instance, CTTS-MM remarkably outperforms Self-consistency by 7.68% and Best of N by 4.83%. Moreover, by only utilizing open-source models, CTTS surpasses flagship closed-source LLMs, including GPT-4.1 and Claude-3.7-sonnet, which demonstrates CTTS-MM can fully release the potential of models during inference time. Our contribution can be summarized as follows:

- We take the first step towards formalizing and analyzing different paradigms of Collective Test-Time Scaling (CTTS), including 1) single agent to multiple reward models (SA-MR); 2) Multiple agents to Single reward model (MA-SR); and 3) multiple agents to multiple reward models(MA-MR). Our study reveals that MA-MR is the optimal CTTS paradigm due to the intra- and inter-collaboration of model groups.

- We propose a novel CTTS framework named CTTS-MM, which combines multiple LLM agents and multiple reward models in a unified search–reward–search pipeline. Specifically, Agent Collaboration Search (ACS) is proposed to dynamically select an optimal combination of agents from a candidate pool, and Mixture of Reward models (MoR) is proposed to provide high-quality feedback. To achieve MoR, a Prior Reward model Ensemble Selection (PRES) with a Pair-wise Reward Ranking (PRR) metric is designed to construct an effective and adaptive reward signal.

- Extensive experiments across multiple benchmarks demonstrate that our CTTS-MM consistently outperforms existing STTS methods (+6.02% over Symbolic-MoE, +7.09% over MoA), leading proprietary LLMs (+7.06% over GPT-4.1) and various open-source LLMs, validating the effectiveness of the proposed CTTS-MM framework and highlighting the substantial potential of collective test time scaling.

## 2 RELATED WORK

**Test-Time Scaling**    Test-time scaling methods Snell et al. (2025); Brown et al. (2024); Madaan et al. (2023); Wang et al. (2023); Du et al. (2024); Wei et al. (2022); Yao et al. (2023); Chen et al. (2024e) mainly focus on how to enhance LLM agents' capabilities at test time. Best of N Snell et al. (2025) is a classic TTS approach that generate answers multiple times with LLM agents and obtains the best answer based on the reward score. Similar methods Chen et al. (2024e) called self-consistency essentially follow the same paradigm, except that they use a verifier to select the answer. This verifier can be an evaluation tool or an algorithm like majority voting. Self-refine Madaan et al. (2023) obtains the optimal solution through a self-evaluation and self-correction approach while Du et al. (2024) employs a multi-round debating between two agents to reach the final answer.

**Multi-agent Collaboration**    A growing number of researches have explored collaborative strategies among multiple agents. Emerging research Chen et al. (2025); Lu et al. (2024); Shnitzer et al. (2024); Srivatsa et al. (2024); Wang et al. (2025) aims to make selection decisions before response generation, directing queries to appropriate agents in advance. MoA Wang et al. (2025) exemplifies this by assigning LLM agents into an ensemble system. Symbolic-MoE Chen et al. (2025) proposes a Mixture-of-Experts framework that dynamically selects and combines agents based on skill-specific expertise. Other methods Chen et al. (2024c); junyou li et al. (2024); Chen et al. (2024d); Gui et al. (2024); Wang et al. (2023) fuse the results of multiple model outputs to yield a refined answer.

## 3 METHODOLOGY

In this section, we first provide a brief preliminary to elaborate on the specific framework of three CTTS paradigms. Then we introduce our proposed CTTS-MM. In Section 3.2, we introduce our Agent Collaboration Search (ACS). Section 3.3 details our proposed Mixture of Reward model (MoR) for selecting the optimal combination of reward models. The construction of a question pool for later selection is first presented. We then introduce Pair-wise Reward Ranking (PRR) and Prior Reward Model Ensemble Selection (PRES). Overall framework is illustrated in Figure 3

### 3.1 PRELIMINARY

Figure 1 illustrates three CTTS paradigms we aim to explore: (1) single agent to multiple reward models (SA-MR); (2) multiple agents to single reward model (MA-SR); (3) multiple agents to multiple reward models (MA-MR). We design a search-reward framework to systematically investigate all three paradigms. For the specific framework setting of each paradigm, MA-SR performs multi-agent ACS with a single reward model while SA-MR adopts ACS using a single agent with MoR. Note that for SA-MR, ACS is performed under multiple answers generated by a single agent. Finally, MA-MR builds upon the previous two paradigms by jointly leveraging ACS and MoR.

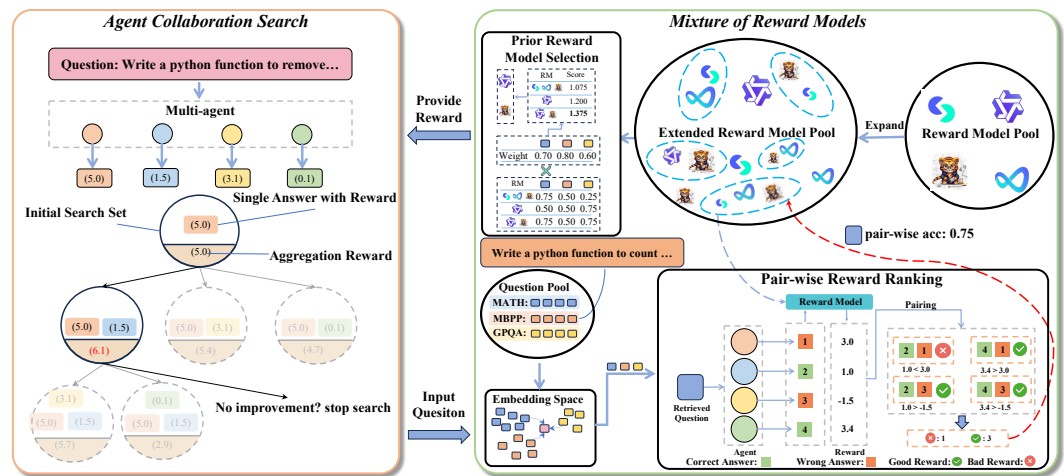

Figure 3: Overview of the proposed CTTS-MM framework. The left part illustrates the Agent Collaboration Search (ACS) while the right part depicts the Mixture of Reward Models (MoR).

## 3.2 AGENT COLLABORATION SEARCH

The process of ACS is illustrated in Figure 3. We design ACS based on a simple yet effective greedy search algorithm with early stop and residual aggregation. Specifically, given a question $q$, we first collect $n$ candidate answers from $n$ agents (under SA-MR setting, $n$ candidates come from repeated generation of one agent), denoted as $\mathcal{A} = \{A_0, A_1, \ldots, A_{n-1}\}$. Our goal is to obtain the optimal answer from these candidates through an iterative and reward-guided greedy search. We begin by computing the reward score for each candidate using Mixture of Reward Models, denoted as function MoR. Specifically, for each answer $A_i$, we obtain its reward score by:

$$r_i = \text{MoR}(q, A_i), \quad i = 0, 1, \ldots, n-1. \tag{1}$$

We then sort the candidates based on their scores and select the top-$k$ answers to initialize our search set $\mathcal{S}^{(0)}$:

$$\mathcal{S}^{(0)} = \{A_{(0)}, A_{(1)}, \ldots, A_{(k-1)}\}, \tag{2}$$

where $A_{(i)}$ denotes the $i$-th ranked answer by score. An aggregator $Agg$ is then used to summarize the current set of answers into a single composite response:

$$C^{opt} = \text{Agg}(\mathcal{S}^{(0)}), \tag{3}$$

and its corresponding reward score is computed as:

$$r^{opt} = \text{MoR}(q, C^{opt}). \tag{4}$$

where $C^{opt}$ and $r^{opt}$ are the current optimal answer and its corresponding reward score, respectively.

We then iteratively check whether augmenting the initial search set $\mathcal{S}^{(0)}$ with a remaining candidate $A_j \in \mathcal{A} \setminus \mathcal{S}^{(0)}$ can yield a better answer. For each such candidate $A_j$, we compute:

$$\hat{C}_j = \text{Agg}(\mathcal{S}^{(0)} \cup \{A_j\}), \tag{5}$$

$$\hat{r}_j = \text{MoR}(q, \hat{A}_j). \tag{6}$$

We identify the candidate $A_j^*$ that yields the highest reward score:

$$A_j^* = \arg\max_{A_j \in \mathcal{A} \setminus \mathcal{S}^{(0)}} \hat{r}_j. \tag{7}$$

If $\hat{r}_{j^*} > r^{opt}$, we update:

$$\begin{cases} \mathcal{S}^{(1)} = \mathcal{S}^{(0)} \cup \{A_j^*\}, \\ C^{opt} = \hat{C}_{j^*}, \\ r^{opt} = \hat{r}_{j^*} \end{cases} \tag{8}$$

and repeat the process using $\mathcal{S}^{(1)}$ as the new base set. Otherwise, if no such improvement is found, the search terminates and $C^{opt}$ is taken as the current optimal answer. Moreover, to mitigate potential information loss during greedy search, we incorporate a residual aggregation step. Specifically, we aggregate the final optimal answer $C^{opt}$ with the initial candidate set $\mathcal{A}$ to produce:

$$\begin{cases} C^{\text{res}} = \text{Agg}(\mathcal{A} \cup \{C^{opt}\}), \\ r^{\text{res}} = \text{MoR}(q, C^{\text{res}}) \end{cases} \tag{9}$$

If $r^{\text{res}} > r^{opt}$, we replace $C^{opt}$ with $C^{\text{res}}$ as the final output. Otherwise, we keep the original output.

### 3.3 MIXTURE OF REWARD MODEL

The multi-reward-model system aims to provide accurate reward scores for the preceding greedy search process. The key challenge lies in selecting suitable reward models for different questions since reward models are currently very domain-specific. Existing approaches Snell et al. (2025) typically rely on manually selecting specific reward models tailored to specific datasets or domains. While such methods may perform well on particular datasets, they lack generalization and flexibility. We argue that this challenge is fundamentally aligned with the motivation behind MoA, which aims to enhance performance and generalization through complementary collaboration among diverse agents. The essence of MoR is somewhat analogous, which is enhancing the precision of the provided rewards through the interaction and collaboration of reward models across different domains. Our core idea is to first expand the individual reward models by constructing a reward model pool and combining them through different subsets of reward models using various weighting methods. This approach allows us to significantly extend the capacity of the original pool. The next step is to select the most suitable individual reward models or their weighted combinations from the pool. Inspired by Retrieval-Augmented Generation (RAG) methods Lewis et al. (2020); Chen et al. (2024a), we introduce a diversified question pool as a prior for selecting the best reward model(s). We then propose a novel ranking metric called Pair-wise Reward Ranking (PRR) to evaluate the capability of reward models in assessing outputs from LLM agents. Based on the above techniques, a prior based reward model selection method named Prior Reward Model Ensemble Selection (PRES) is proposed. In this section, we first describe how the question pool is constructed, followed by the introduction of PRR. Finally, the details of PRES are presented.

#### 3.3.1 DIVERSIFIED QUESTION POOL

As mentioned, manual selection of reward model(s) based on the domain of the dataset is neither generalizable nor flexible. On the other hand, it is difficult to directly select reward model(s) based on their architectures or parameters. To address this, we introduce a diversified question pool as a form of prior knowledge to guide the selection process. We construct the question pool using the validation sets of diverse tasks, such as math reasoning and coding. Then, for each question in the pool, we evaluate the correctness of each LLM agent's response, which serves as prior knowledge for the subsequent selection process.

#### 3.3.2 PAIR-WISE REWARD RANKING

Given the constructed question pool $\mathcal{Q} = \{q_1, q_2, \ldots, q_N\}$, we aim to evaluate whether the reward score provided by the reward models is accurate. That is to say, for the same question, correct answers should receive higher scores than incorrect ones. Specifically, given a question $q \in \mathcal{Q}$, let $\mathcal{A}_q = \{(a_1, y_1), (a_2, y_2), \ldots, (a_n, y_n)\}$ denote the set of answers provided by $n$ agents, where $a_i$ is the answer generated by the $i$-th agent and $y_i \in \{0, 1\}$ indicates whether the answer is correct (1) or incorrect (0). $\mathcal{A}_q$ is then partition into two subsets:

$$\begin{cases} \mathcal{A}_q^{\text{pos}} = \{a_i \mid y_i = 1\}, \\ \mathcal{A}_q^{\text{neg}} = \{a_j \mid y_j = 0\}, \end{cases} \tag{10}$$

where $\mathcal{A}_q^{\text{pos}}$ and $\mathcal{A}_q^{\text{neg}}$ represent correct and incorrect responses, respectively. We then construct all possible pairs $(a_i, a_j)$ where $a_i \in \mathcal{A}_q^{\text{pos}}$ and $a_j \in \mathcal{A}_q^{\text{neg}}$. For each pair, we query the reward function $MoR(\cdot)$ to obtain their reward scores, denoted as $r(a_i) = MoR(a_i, q)$ and $r(a_j) = MoR(a_j, q)$. If

$r(a_i) > r(a_j)$, we consider this pair to be accurately assessed by the reward model. The pair-wise accuracy of the reward model on question $q$ can then be defined as:

$$\text{Acc}_q = \frac{1}{|\mathcal{P}_q|} \sum_{(a_i, a_j) \in \mathcal{P}_q} I[r(a_i) > r(a_j)], \tag{11}$$

where $\mathcal{P}_q$ is the set of all valid answer pairs for $q$, and $I[\cdot]$ is the indicator function. We can then rank the entire reward model pool on a given question q using $\text{Acc}_q$, which serves as the criterion for subsequent reward model selection. Besides, for questions where all agents provide either entirely correct or entirely incorrect answers, we consider them invalid, as the reward model's accuracy cannot be evaluated on such questions. For multiple reward models $\{R_1, R_2, \ldots, R_K\}$, the final reward score for a candidate answer is computed as a weighted combination of the individual reward scores from these $K$ models. Specifically, for a given answer $a$, the reward score from multiple agents is defined as:

$$r_{\text{MoR}}(a, q) = \sum_{k=1}^{K} w_k \cdot R_k(a, q), \tag{12}$$

where $w_k$ denotes the weight assigned to reward model $R_k$. The choice of weight computation plays a crucial role in the effectiveness of the MoR. In this work, we basically utilize three weighting strategies based on the individual reward model accuracies $\{\alpha_1, \alpha_2, \ldots, \alpha_K\}$ obtained by PRR. For Linear weighting, the weight is proportional to the accuracy:

$$w_k = \frac{\alpha_k}{\sum_{j=1}^{K} \alpha_j}. \tag{13}$$

For Softmax weighting, we compute the weights via a softmax by:

$$w_k = \frac{\exp(\alpha_k/\tau)}{\sum_{j=1}^{K} \exp(\alpha_j/\tau)}, \tag{14}$$

where $\tau > 0$ is a temperature parameter. For naive sum, all reward models are treated equally and no weighting is applied. This corresponds to setting $w_k = 1$ for all $k$.

### 3.3.3 PRIOR REWARD MODEL ENSEMBLE SELECTION

Given a question as $q$, a pre-trained embedding model is utilized to embed it into a $d$-dimensional semantic space, resulting in vector $\mathbf{e}_q \in \mathbb{R}^d$. Similarly, the question pool $\mathcal{Q} = \{q_1, \ldots, q_N\}$ can be embedded into a matrix $\mathbf{E} \in \mathbb{R}^{N \times d}$, where each row $\mathbf{e}_i$ is the embedding of $q_i$. We then compute the cosine similarity vector $\mathbf{s} \in \mathbb{R}^N$ by $\mathbf{s} = e_q \cdot \mathbf{E}^T$. We select the top-$k$ questions with the highest similarity scores, forming index set $\mathcal{I}_{\text{top}} \subset \{1, \ldots, N\}$. For each reward model or combination $R$, we retrieve its pair-wise accuracy vector $\boldsymbol{\alpha}^m \in \mathbb{R}^N$ over the top-k question set. Using the selected indices $\mathcal{I}_{\text{top}}$, we compute a final score by weighted dot product:

$$\text{Score}_q^m = \sum_{i \in \mathcal{I}_{\text{top}}} s_i \cdot \alpha_i^m. $$

The final reward model(s) selected for $q$ is:

$$R_q^* = \arg \max_m \text{Score}_q^m. $$

We then use the selected reward model(s) for greedy search.

## 4 EXPERIMENT

In this section, we first analyze exploratory experiments among different CTTS paradigms. Then we present a comprehensive comparison between our CTTS-MM and existing methods across seven benchmark datasets. Finally, we perform a series of analytical and ablation studies to further investigate the effectiveness of our approach.

### 4.1 EXPERIMENTAL SETTING

**Datasets.** To ensure comprehensive evaluation across diverse capabilities, seven multi-domain datasets across four representative task types are utilized: (1) mathematical reasoning (MATH Hendrycks et al. (2021), AIME2024 MAA (2024)); (2) complex knowledge-based reasoning (GPQA Rein et al. (2024)); (3) instruction-following tasks (IFEval Zhou et al. (2023)); (4) code generation (MBPP Austin et al. (2021), LiveCodeBench Jain et al. (2024), HumanEval Mark Chen (2021)). All datasets except HumanEval are partitioned into non-overlapping validation and test subsets. Validation portions are utilized to construct the question pool. Details are provided in Section B of our Appendix.

Table 1: Comparison results of different TTS paradigms. MR* means utilizing our proposed MoR to select reward models. Multi-agent* means utilizing ten chosen LLMs.

| Setting | Model | Reward Model | Weight Method | MBPP | MATH-500 | GPQA | Avg. |
|---|---|---|---|---|---|---|---|
| Single Agent | Qwen2.5-32B-Instruct | - | - | 76.00 | 75.60 | 40.91 | 64.17 |
| | Qwen2.5-72b-Instruct | - | - | 75.80 | 78.80 | 45.45 | 66.68 |
| | Llama-3.3-Nemotron-Super-49B-v1 | - | - | 65.40 | 75.20 | 48.48 | 63.03 |
| SA-SR | Qwen2.5-32B-Instruct | AceCodeRM-32B | - | 77.40 | 78.2 | 47.47 | 67.69 |
| | Qwen2.5-32B-Instruct | Qwen2.5-Math-RM-72B | - | 77.00 | 78.80 | 46.46 | 67.42 |
| | Qwen2.5-72b-Instruct | AceCodeRM-32B | - | 76.60 | 80.20 | 51.01 | 69.27 |
| | Qwen2.5-72b-Instruct | Qwen2.5-Math-RM-72B | - | 76.00 | 80.80 | 50.51 | 69.10 |
| | Llama-3.3-Nemotron-Super-49B-v1 | AceCodeRM-32B | - | 66.40 | 76.00 | 50.80 | 64.40 |
| | Llama-3.3-Nemotron-Super-49B-v1 | Qwen2.5-Math-RM-72B | - | 65.80 | 76.00 | 50.00 | 64.20 |
| SA-MR | Qwen2.5-32B-Instruct | AceCodeRM-32B+Qwen2.5-Math-RM-72B | softmax | 76.6 | 78.2 | 48.48 | 67.76 |
| | Qwen2.5-32b-Instruct | MR* | - | 78.00 | 79.4 | 51.01 | 69.47 |
| | Qwen2.5-72b-Instruct | AceCodeRM-32B+Qwen2.5-Math-RM-72B | softmax | 76.8 | 80.20 | 51.51 | 69.50 |
| | Qwen2.5-72b-Instruct | MR* | - | 77.20 | 81.4 | 53.53 | 70.71 |
| | Llama-3.3-Nemotron-Super-49B-v1 | AceCodeRM-32B+Qwen2.5-Math-RM-72B | softmax | 66.20 | 76.60 | 51.52 | 64.77 |
| | Llama-3.3-Nemotron-Super-49B-v1 | MR* | - | 66.80 | 76.80 | 54.55 | 66.05 |
| MA-SR | Multi-agent* | Skywork-Reward-V2-Llama-3.1-8B-40M | - | 77.00 | 91.20 | 61.11 | 75.97 |
| | Multi-agent* | Qwen2.5-Math-RM-72B | - | 80.6 | 91.8 | 61.11 | 77.84 |
| | Multi-agent* | AceCodeRM-32B | - | 82.2 | 90.8 | 61.62 | 78.21 |
| MA-MR (Proposed CTTS-MM) | Multi-agent* | MR* | - | **83.20** | **93.00** | **64.14** | **80.11** |

Table 2: Main Results of CTTS-MM compared with leading LLMs and other related methods on seven mainstream benchmarks.

| Model | AIME-2024 | MATH-500 | MBPP | LiveCodeBench | GPQA-Diamond | Human-eval | IFEval | Avg |
|---|---|---|---|---|---|---|---|---|
| *Open-source LLMs* | | | | | | | | |
| Qwen-2.5-72B-Instruct | 16.70 | 78.80 | 75.80 | 26.10 | 45.45 | 78.66 | 86.30 | 58.26 |
| DeepSeek-R1-Distill-Llama-70B | 60.00 | 82.80 | 76.40 | 40.70 | 60.10 | 92.07 | 80.30 | 70.34 |
| Llama-3.3-Nemotron-Super-49B-v1 | 16.70 | 75.20 | 65.40 | 28.00 | 48.48 | 84.76 | 82.70 | 57.32 |
| QwQ-32B | 46.70 | 87.80 | 81.80 | 38.60 | 57.07 | 92.07 | 81.70 | 69.39 |
| InternLM2.5-20B-Chat | 3.30 | 55.20 | 55.00 | 14.90 | 34.85 | 69.51 | 64.70 | 42.49 |
| Gemma-3-27b-it | 30.00 | 84.00 | 70.40 | 27.70 | 50.51 | 86.59 | 81.00 | 61.46 |
| Qwen2.5-32b-Instruct | 20.00 | 75.60 | 76.00 | 24.00 | 40.91 | 77.44 | 78.70 | 56.09 |
| TeleChat2-35B-32K | 10.00 | 70.00 | 70.00 | 19.50 | 33.33 | 73.17 | 82.00 | 51.14 |
| EXAONE-Deep-32B | 33.30 | 84.38 | 72.80 | 31.60 | 58.59 | 93.90 | 76.30 | 64.41 |
| GLM-Z1-32B-0414 | 66.70 | 90.00 | 74.40 | 44.40 | 59.60 | 96.34 | 83.00 | 73.49 |
| Llama-3.3-70B-Instruct | 30.00 | 73.00 | 70.40 | 30.10 | 46.97 | 84.15 | 90.00 | 60.66 |
| Qwen3-32B | 53.30 | 88.00 | 50.60 | 33.40 | 65.15 | 90.85 | 83.70 | 66.43 |
| Qwen2.5-Coder-32B-Instruct | 16.70 | 73.60 | 78.00 | 27.70 | 41.92 | 87.80 | 80.30 | 58.00 |
| HuatuoGPT-o1-72B | 16.70 | 73.00 | 78.00 | 27.40 | 50.00 | 85.37 | 74.00 | 57.78 |
| DeepSeek-R1-Distill-Qwen-32B | 56.70 | 85.60 | 81.00 | 44.70 | 60.10 | 95.73 | 73.70 | 71.08 |
| *Proprietary LLMs* | | | | | | | | |
| GPT-4.1 (2025-04-14) | 50.00 | 85.80 | 79.20 | 42.20 | 67.17 | 92.07 | 86.00 | 71.78 |
| Claude-3.7-Sonnet (2025-02-19) | 26.70 | 73.20 | 75.40 | 41.30 | 63.64 | 90.85 | 88.00 | 65.58 |
| GPT-4o (2024-08-06) | 10.00 | 74.60 | 74.20 | 29.80 | 52.53 | 85.36 | 82.30 | 58.40 |
| Claude-3.5-Sonnet (2024-06-20) | 16.70 | 74.20 | 75.80 | 34.30 | 61.62 | 89.63 | 80.30 | 61.79 |
| *Related Methods* | | | | | | | | |
| Majority Voting Chen et al. (2024c) | 56.67 | 90.20 | 80.40 | 34.65 | 26.26 | 89.63 | 80.67 | 65.50 |
| Multi-Agent Verification Lifshitz et al. (2025) | 63.33 | 76.30 | 74.60 | 42.55 | 59.00 | 92.00 | 83.00 | 70.11 |
| Symbolic-MoE Chen et al. (2025) | 50.00 | 90.40 | 82.60 | 43.16 | 62.63 | 92.07 | 89.00 | 72.82 |
| MoA Wang et al. (2025) | 53.33 | 87.80 | 82.00 | 40.12 | 58.80 | 90.85 | 89.33 | 71.75 |
| Self Consistency Chen et al. (2024e) | 70.00 | 91.40 | 82.40 | 30.47 | 65.15 | 90.39 | 68.33 | 71.16 |
| Best of N Snell et al. (2025) | 66.67 | 90.8 | 75.00 | 44.98 | 60.61 | 96.34 | 83.66 | 74.01 |
| *Ours v.s. Strong Baselines* | | | | | | | | |
| **CTTS-MM(ours)** | 70.00 | 93.00 | 83.20 | 52.28 | 64.14 | 97.56 | 91.67 | 78.84 |
| *- v.s. GLM-Z1-32B-0414* | ↑3.30 | ↑3.00 | ↑8.80 | ↑7.88 | ↑4.54 | ↑1.22 | ↑8.67 | ↑5.34 |
| *- v.s. GPT-4.1* | ↑20.00 | ↑7.20 | ↑4.00 | ↑10.08 | ↓3.03 | ↑5.49 | ↑5.67 | ↑7.06 |
| *- v.s. Best of N* | ↑3.30 | ↑2.20 | ↑8.20 | ↑7.29 | ↑3.53 | ↑1.22 | ↑8.01 | ↑4.82 |

**LLM agents and Reward Models.** For LLM agents used in our experiments, we assemble a set of ten mid-sized open-source LLMs (ranging from 20B to 72B parameters) from diverse architecture families. For reward models, we select eight off-the-shelf models, covering specialized domains like math and coding. Additional details are provided in the Appendix.

Table 3: Component ablation on four standard datasets.

| MoR | ACS | Residual Aggregation | MATH-500 | MBPP | AIME | LiveCodeBench |
|-----|-----|----------------------|----------|------|------|---------------|
| ✗ | ✗ | ✗ | 90.80 | 80.00 | 56.67 | 40.12 |
| ✗ | ✓ | ✗ | 91.20 | 80.20 | 60.00 | 43.16 |
| ✓ | ✗ | ✗ | 91.40 | 80.20 | 63.33 | 43.77 |
| ✗ | ✓ | ✓ | 91.80 | 80.60 | 66.67 | 44.38 |
| ✓ | ✓ | ✗ | 92.40 | 83.00 | 70.00 | 51.67 |
| ✓ | ✓ | ✓ | 93.00 | 83.20 | 70.00 | 52.28 |

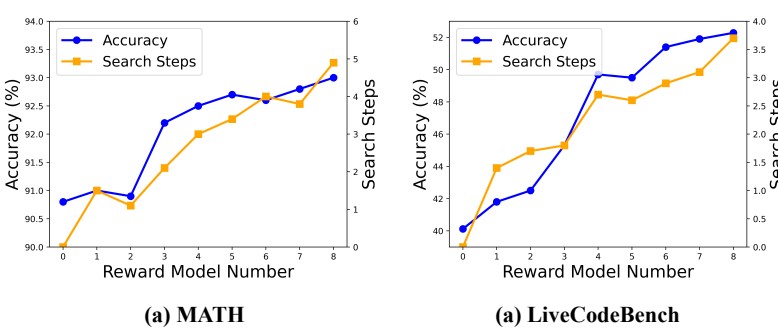

**(a) MATH**      **(a) LiveCodeBench**

Figure 4: The curve of scaling capability of Reward Models.

## 4.2 ANALYSIS ON DIFFERENT TTS PARADIGMS

To thoroughly investigate CTTS and STTS paradigms, we conduct exploratory experiments measuring performance variation among different paradigms. As demonstrated in Table 1, CTTS paradigms outperform both STTS and single-model baselines under most settings. For example, under MA-MR setting, our method achieves an average improvement of 10.84% (80.11% v.s. 69.27%) over the best performance in the SA-SR setting, with gains of 5.8% (83.20% v.s. 77.40%) on MBPP, 12.20% (93.00% v.s. 80.80%) on MATH, and 13.13% (64.14 vs. 51.01) on GPQA, respectively. Under MA-SR setting, the best performance shows an average improvement of 8.94% over SA-SR while SA-MR yields an average gain of 1.44%. The results indicate the effectiveness of our proposed CTTS paradigm. In particular, our MA-MR based framework achieves superior improvements. Besides, we observe that under the SA-MR setting, a fixed combination of reward models may lead to performance degradation compared to SA-SR. This indicates that naively combining multiple reward models is unlikely to yield improvements and can even bring performance dropping. Such results are expected since most reward models are domain-specific. Fixed combination cannot guarantee consistent gains across all problems from different domains, which underscores the necessity of our proposed MoR for adaptive reward model selection. Comparative results between MA-MR and MA-SR also demonstrate that multi-RM achieves superior generalization compared to domain-specific reward models. Refer to our Appendix for more results.

## 4.3 MAIN RESULTS

As demonstrated in Table 2, our proposed CTTS-MM demonstrates significant improvements across seven diverse benchmarks. Through comprehensive comparisons with (1) fifteen representative open-source models (2) four leading Proprietary models and (3) five existing collaboration methods, our approach demonstrates consistent and substantial improvements across all evaluation dimensions. Our framework achieves 78.84% average accuracy on seven benchmarks. Compared to existing collaboration approaches, CTTS-MM outperforms Majority Voting Chen et al. (2024c) by +13.34%, MAV Lifshitz et al. (2025) by +8.73%, Symbolic-MoE Chen et al. (2025) by +6.02%, MoA Wang et al. (2025) by 7.09%, Self Consistency Chen et al. (2024e) by 7.68% and Best of N Snell et al. (2025) by 4.83%. Remarkably, our approach demonstrates superior performance compared to strong baselines from open-source LLM agents, proprietary LLM agents, and related multi-agent methods. Specifically, on average accuracy, CTTS-MM surpasses the best-performing open-source LLM GLM-Z1-32B by 5.34%, proprietary LLM GPT-4.1 by 7.06%, and STTS-based method Best-of-N by 4.82%. These results demonstrate that our CTTS-MM can effectively leverage the complementary advantages of multiple agents and multiple reward models, leading to a superior performance increase. This further validates the substantial potential of the CTTS paradigm, par-

Table 4: Comparative results of average inference time on MATH-500 and LiveCodeBench.

| Method | MATH-500 | | LiveCodeBench | |
|---|---|---|---|---|
| | Inference Time (s) | Accuracy (%) | Inference Time (s) | Accuracy (%) |
| Multi-Agent Verification | 22.45 | 76.30 | 24.52 | 42.55 |
| Symbolic-MoE | 15.05 | 90.40 | 17.76 | 43.16 |
| MoA | 19.54 | 87.80 | 21.97 | 40.12 |
| Best of N | 13.12 | 90.80 | 16.17 | 44.98 |
| CTTS-MM (Ours) | 20.14 | 93.00 | 22.04 | 52.28 |

ticularly highlighting the MA-MR framework's robust capability in multiple domains. Refer to our Appendix for more results.

## 4.4 ABLATION STUDY

We perform a comprehensive component-wise ablation study on four standard benchmarks to quantify the contribution of each component in our CTTS-MM framework. Note that Residual Aggregation can only be applied when Agent Collaboration Search (ACS) is utilized. As illustrated in Table 3, the baseline obtains 90.08% accuracy on MATH. Utilizing ACS and MoR improves performance by 0.4% and 0.6%, respectively, reaching 92.40% when combined. Further gains come from Residual Aggregation, which contributes an additional 0.6%. Similar improvements are observed on the other three benchmarks, indicating the effectiveness of each component in CTTS-MM.

## 4.5 ANALYSIS ON SCALING CAPABILITY OF RMS

To investigate the scalability of MoR, we conduct experiments measuring performance improvements with increasing numbers of reward models. As shown in Figure 4, the performance of CTTS-MM consistently improves with increasing number of reward models on both MATH and LiveCodeBench. For instance, on LiveCodeBench, CTTS-MM achieves an accuracy of approximately 41.8% with a single reward model. When the number of reward models increases to four, the accuracy improves to nearly 50%, and ultimately reaches 52.28% with all eight reward models. Moreover, we observe that the search step also gradually increases with more reward models. It indicates that with the increase of reward models, our Mixture of Reward Model can enhance the entire model pool to extract cross-domain information, leading to more accurate reward scores for question-answer pairs during search. It guides the search process in a more optimal direction. In contrast, when the RM pool is limited in size, its robustness is weaker, resulting in inaccurate reward scores. This can mislead the search direction, potentially causing early stop and local optimal solutions.

## 4.6 ANALYSIS ON TIME AND COST EFFICIENCY

Although we primarily focus on exploring the optimal paradigm under collective test-time scaling, we further investigate the inference time and cost efficiency of our CTTS-MM in comparison to other related methods. For time efficiency, we report average inference time per response on MATH-500 and LiveCodeBench. As shown in Table 4, CTTS-MM achieves a significant improvement in accuracy with bearable inference time compared to other methods. For cost efficiency, we plot results of performance against inference cost. Refer to Section E of our Appendix for the detailed figure.

## 5 CONCLUSION

In this manuscript, we first explore Collective Test-Time Scaling (CTTS). We propose and investigate three CTTS paradigms: SA-MR, MA-SR and MA-MR. Experiments demonstrate that CTTS outperforms previous Single TTS paradigms, while the MA-MR variant consistently achieves superior performance. Based on it, we further propose a CTTS framework called CTTS-MM. To search for optimal agent ensembles, we propose Agent Collaboration Search approach. For adaptively selecting multiple reward models, Prior Reward Models Ensemble Selection is proposed. Experiments on seven benchmarks verify the superiority of CTTS-MM, revealing the strong potential of CTTS.

## ETHICS STATEMENT

All authors have read and adhered to the ICLR Code of Ethics. This work involves no human subjects, sensitive personal data, or potentially harmful applications. All datasets used in our experiments are publicly available from official sources, and no proprietary or private data are involved. The authors declare that there are no potential conflicts of interest related to this work.

## REPRODUCIBILITY STATEMENT

To support reproducibility, we provide a complete anonymous codebase in the supplementary material. The repository contains the implementation of our proposed CTTS-MM, the evaluation pipeline, and an associate example dataset, along with a README that provides installation and usage instructions. Besides, in our Appendix, Section B contains more details on our experiment datasets, Section C contains more details on LLMs and reward models we use for the experiment while Section D contains our implementation details. Together, these resources enable reliable reproduction of our experiments and results.

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

APPENDIX

This supplementary document is organized as follows:

- Section A contains details on our use of Large Language Models.
- Section B contains more details on our experiment datasets.
- Section C contains more details on LLMs and reward models we use for the experiment.
- Section D contains our implementation details.
- Section E contains experiment results on cost efficiency.
- Section F contains more comparison experiment results.
- Section G contains results for the analysis on specific cases from our approaches.
- Section H contains experiment results for the analysis on question pool.
- Section I contains more details on our Agent Collaboration Search.
- Section J contains details on our prompts for each of the seven benchmarks.

## A  DETAILS ON THE USE OF f LARGE LANGUAGE MODELS

In this paper, we employ a large language model (LLM) to assist our writing, primarily for aiding or polishing the paper and no other applications are included. Specifically, we use DeepSeek of chat-version via this website: https://yuanbao.tencent.com/chat. Our usage of the LLM is limited to the following purposes: (1) translating terms and sentences, and (2) refining the phrasing of the manuscript. Finally, we acknowledge the convenient user-interactive LLM service provided by Tencent Yuan Bao based on DeepSeek.

## B  DETAILS ON DATASET

In our experiments, we assess the effectiveness of our proposed CTTS-MM across seven diverse benchmarks covering mathematical reasoning, complex QA, instruction following, and code generation. Note that, except for HumanEval Mark Chen (2021), all datasets are split into test and validation sets, with the validation sets utilized to construct the question pool. For MBPP Austin et al. (2021), we retain the original test set and merge the training and validation sets to serve as the validation split. Specifically, the validation set consists of 464 samples while the test set contains 500 samples. For LiveCodeBench Jain et al. (2024), we utilize their v5 version as the test set, reserving v6 for validation. For MATH Hendrycks et al. (2021), we evaluate on the MATH-500 subset and randomly sample 1,000 samples from the original dataset for validation. For AIME MAA (2024), we use the 2024 competition problems as the test set, leveraging historical questions (1983–2023) for validation. For GPQA Rein et al. (2024), we adopt the diamond subset consisting of graduate-level science questions as the test set, with the rest used for validation. In the IFEval Zhou et al. (2023), 300 instruction-following samples are selected at random for testing, with 241 used for validation. Finally, for Human-eval, we simply use their original version for test split (164 samples) and no validation split is constructed as mentioned.

## C  DETAILS ON LLMS AND RMS

### C.1  LLM USAGE

As we mentioned in our manuscript, we assemble a set of ten mid-sized open-source LLMs (ranging from 20B to 72B parameters) from diverse architecture families. Specifically, the selected LLMs include: Qwen2.5-32B-Instruct Team (2024a), Qwen-2.5-72B-Instruct Team (2024a), Qwen2.5-Coder-32B-Instruct Hui et al. (2024), GLM-Z1-32B-0414 GLM et al. (2024), DeepSeek-R1-Distill-Qwen-32B DeepSeek-AI (2025), DeepSeek-R1-Distill-Llama-70B DeepSeek-AI (2025), QwQ-32B Team (2024b), InternLM2.5-20B-Chat Cai et al. (2024), Llama-3.3-70B-Instruct Grattafiori et al. (2024), Llama-3.3-Nemotron-Super-49B-v1 Bercovich et al. (2025). Note that this pool of

10 LLMs primarily acts as multi-agent in our CTTS-MM framework and is utilized for comparative experiments on TTS paradigms. For comparison experiments against other methods, we additionally include five open-source models: Gemma-3-27b-it Team et al. (2024), TeleChat2-35B-32K Wang et al. (2024b), EXAONE-Deep-32B LG AI Research (2025), Qwen3-32B Team (2025), HuatuoGPT-o1-72B Chen et al. (2024b). Details are listed in Table E.

Table E: Details on utilized LLMs.

| Name | Size | Type |
|---|---|---|
| TeleChat2-35B-32K | 35B | Instruction-tuned |
| GLM-Z1-32B-0414 | 32B | Deep Thinking |
| Qwen-2.5-72B-Instruct | 72B | Instruction-tuned |
| Llama-3.3-70B-Instruct | 70B | Instruction-tuned |
| DeepSeek-R1-Distill-Llama-70B | 70B | Deep Thinking |
| DeepSeek-R1-Distill-Qwen-32B | 32B | Deep Thinking |
| Gemma-3-27b-it | 27B | Instruction-tuned |
| Qwen2.5-Coder-32B-Instruct | 32B | Instruction-tuned |
| Qwen3-32B | 32B | Deep Thinking |
| Llama-3.3-Nemotron-Super-49B-v1 | 49B | Deep Thinking |
| Qwen2.5-32B-Instruct | 32B | Instruction-tuned |
| QwQ-32B | 32B | Deep Thinking |
| EXAONE-Deep-32B | 32B | Deep Thinking |
| HuatuoGPT-o1-72B | 72B | Deep Thinking |
| InternLM2.5-20B-Chat | 20B | Instruction-tuned |

## C.2 RM USAGE

We collect eight off-the-shelf reward models for all our experiments. Specifically, the collected reward models include: Qwen2.5-Math-RM-72B Yang et al. (2024a), Qwen2.5-Math-PRM-7B Zhang et al. (2025), Skywork-Reward-Gemma-2-27B Liu et al. (2024), INF-ORM-Llama3.1-70B Wang et al. (2024a), LDL-Reward-Gemma-2-27B-v0.1 Chen (2025), AceCodeRM-32B Zeng et al. (2025), QRM-Gemma-2-27B Dorka (2024), Skywork-Reward-V2-Llama-3.1-8B-40M Liu et al. (2025). Details are listed in Table F

Table F: Details on utilized Reward Models.

| Name | Size | Base Model | type |
|---|---|---|---|
| Qwen2.5-Math-RM-72B | 72B | Qwen2.5-Math-72B | ORM |
| Qwen2.5-Math-PRM-7B | 7B | Qwen2.5-Math-7B-Instruct | PRM |
| Skywork-Reward-Gemma-2-27B | 27B | Gemma-2-27B-it | ORM |
| INF-ORM-Llama3.1-70B | 70B | Llama-3.1-70B-Instruct | ORM |
| LDL-Reward-Gemma-2-27B-v0.1 | 27B | Gemma-2-27B-it | ORM |
| AceCodeRM-32B | 32B | Qwen2.5-Coder-32B-Instruct | ORM |
| QRM-Gemma-2-27B | 32B | Gemma-2-27B-it | ORM |
| Skywork-Reward-V2-Llama-3.1-8B-40M | 8B | Llama-3.1-8B-Instruct | ORM |

## D  IMPLEMENTATION DETAILS

### D.1  INFERENCE DETAILS

All experiments are conducted under the same inference settings. We employ VLLM Kwon et al. (2023) as the backend for executing LLM inference. The sampling temperature is fixed at 0.7, and the output sequence is set to 8,192 tokens to prevent excessively long generations. A presence penalty of 1.05 is applied to discourage repetitive outputs. In cases where the input context exceeds the model's token limit, we apply the YaRN method Peng et al. (2023) to extend the context

Table G: Comparison results of different TTS paradigms on MBPP, MATH-500 and GPQA. MR* means utilizing our proposed MoR to select reward models. Multi-agent* means utilizing ten chosen LLMs.

| Setting | Model | Reward Model | Weight Method | MBPP | MATH-500 | GPQA | Avg. |
|---|---|---|---|---|---|---|---|
| Single Agent | Qwen2.5-32B-Instruct | - | - | 76.00 | 75.60 | 40.91 | 64.17 |
| | Qwen2.5-72b-Instruct | - | - | 75.80 | 78.80 | 45.45 | 66.68 |
| | Llama-3.3-Nemotron-Super-49B-v1 | - | - | 65.40 | 75.20 | 48.48 | 63.03 |
| | Llama-3.3-70B-Instruct | - | - | 70.40 | 73.00 | 46.97 | 63.46 |
| | DeepSeek-R1-Distill-Llama-70B | - | - | 76.40 | 82.8 | 60.10 | 73.10 |
| SA-SR | Qwen2.5-32B-Instruct | AceCodeRM-32B | - | 77.40 | 78.2 | 47.47 | 67.69 |
| | Qwen2.5-32B-Instruct | Qwen2.5-Math-RM-72B | - | 77.00 | 78.80 | 46.46 | 67.42 |
| | Qwen2.5-72b-Instruct | AceCodeRM-32B | - | 76.60 | 80.20 | 51.01 | 69.27 |
| | Qwen2.5-72b-Instruct | Qwen2.5-Math-RM-72B | - | 76.00 | 80.80 | 50.51 | 69.10 |
| | Llama-3.3-Nemotron-Super-49B-v1 | AceCodeRM-32B | - | 66.40 | 76.00 | 50.80 | 64.40 |
| | Llama-3.3-Nemotron-Super-49B-v1 | Qwen2.5-Math-RM-72B | - | 65.80 | 76.80 | 50.00 | 64.20 |
| | Llama-3.3-70B-Instruct | AceCodeRM-32B | - | 71.20 | 73.40 | 48.00 | 64.20 |
| | Llama-3.3-70B-Instruct | Qwen2.5-Math-RM-72B | - | 70.80 | 73.80 | 47.47 | 64.02 |
| | DeepSeek-R1-Distill-Llama-70B | AceCodeRM-32B | - | 77.00 | 82.60 | 59.09 | 72.90 |
| | DeepSeek-R1-Distill-Llama-70B | Qwen2.5-Math-RM-72B | - | 76.60 | 83.20 | 59.09 | 72.96 |
| SA-MR | Qwen2.5-32B-Instruct | AceCodeRM-32B+Qwen2.5-Math-RM-72B | softmax | 76.60 | 78.20 | 48.48 | 67.76 |
| | Qwen2.5-32B-Instruct | AceCodeRM-32B+Qwen2.5-Math-RM-72B | linear | 76.60 | 78.00 | 47.80 | 67.47 |
| | Qwen2.5-32B-Instruct | MR* | - | 78.00 | 79.4 | 51.01 | 69.47 |
| | Qwen2.5-72B-Instruct | AceCodeRM-32B+Qwen2.5-Math-RM-72B | softmax | 76.80 | 80.20 | 51.51 | 69.50 |
| | Qwen2.5-72B-Instruct | AceCodeRM-32B+Qwen2.5-Math-RM-72B | linear | 77.00 | 80.20 | 52.02 | 69.74 |
| | Qwen2.5-72B-Instruct | MR* | - | 77.20 | 81.4 | 53.53 | 70.71 |
| | Llama-3.3-Nemotron-Super-49B-v1 | AceCodeRM-32B+Qwen2.5-Math-RM-72B | softmax | 66.20 | 76.60 | 51.52 | 64.77 |
| | Llama-3.3-Nemotron-Super-49B-v1 | AceCodeRM-32B+Qwen2.5-Math-RM-72B | linear | 66.20 | 76.40 | 51.52 | 64.70 |
| | Llama-3.3-Nemotron-Super-49B-v1 | MR* | - | 66.80 | 76.80 | 54.55 | 66.05 |
| | Llama-3.3-70B-Instruct | AceCodeRM-32B+Qwen2.5-Math-RM-72B | softmax | 71.40 | 74.00 | 48.48 | 64.63 |
| | Llama-3.3-70B-Instruct | AceCodeRM-32B+Qwen2.5-Math-RM-72B | linear | 71.40 | 74.00 | 48.99 | 64.80 |
| | Llama-3.3-70B-Instruct | MR* | - | 72.00 | 74.40 | 49.49 | 65.30 |
| | DeepSeek-R1-Distill-Llama-70B | AceCodeRM-32B+Qwen2.5-Math-RM-72B | softmax | 76.60 | 83.00 | 60.10 | 73.23 |
| | DeepSeek-R1-Distill-Llama-70B | AceCodeRM-32B+Qwen2.5-Math-RM-72B | linear | 76.80 | 83.20 | 60.10 | 73.36 |
| | DeepSeek-R1-Distill-Llama-70B | MR* | - | 77.20 | 83.60 | 60.60 | 73.80 |
| MA-SR | Multi-agent* | Skywork-Reward-V2-Llama-3.1-8B-40M | - | 77.00 | 91.20 | 61.11 | 75.97 |
| | Multi-agent* | Qwen2.5-Math-RM-72B | - | 80.6 | 91.8 | 61.11 | 77.84 |
| | Multi-agent* | LDL-Reward-Gemma-2-27B-v0.1 | - | 78.80 | 91.00 | 62.63 | 77.48 |
| | Multi-agent* | AceCodeRM-32B | - | 82.2 | 90.8 | 61.62 | 78.21 |
| MA-MR (Proposed CTTS-MM) | Multi-agent* | MR* | - | **83.20** | **93.00** | **64.14** | **80.11** |

Table H: Main results of CTTS-MM compared with the optimal setting of Best of N on five benchmarks.

| Model | AIME | MATH-500 | MBPP | LiveCodeBench | Human-eval | Avg |
|---|---|---|---|---|---|---|
| *Open-source LLMs* | | | | | | |
| Qwen-2.5-72B-Instruct | 16.70 | 78.80 | 75.80 | 26.10 | 78.66 | 55.21 |
| DeepSeek-R1-Distill-Llama-70B | 60.00 | 82.80 | 76.40 | 40.70 | 92.07 | 70.39 |
| Llama-3.3-Nemotron-Super-49B-v1 | 16.70 | 75.20 | 65.40 | 28.00 | 84.76 | 54.01 |
| QwQ-32B | 46.70 | 87.80 | 81.80 | 38.60 | 92.07 | 69.39 |
| InternLM2.5-20B-Chat | 3.30 | 55.20 | 55.00 | 14.90 | 69.51 | 39.58 |
| Gemma-3-27b-it | 30.00 | 84.00 | 70.40 | 27.70 | 86.59 | 59.74 |
| Qwen2.5-32b-Instruct | 20.00 | 75.60 | 76.00 | 24.00 | 77.44 | 54.61 |
| TeleChat2-35B-32K | 10.00 | 70.00 | 70.00 | 19.50 | 73.17 | 48.53 |
| EXAONE-Deep-32B | 33.30 | 84.38 | 72.80 | 31.60 | 93.90 | 63.20 |
| GLM-Z1-32B-0414 | 66.70 | 90.00 | 74.40 | 44.40 | 96.34 | 74.37 |
| Llama-3.3-70B-Instruct | 30.00 | 73.00 | 70.40 | 30.10 | 84.15 | 57.53 |
| Qwen3-32B | 53.30 | 88.00 | 50.60 | 33.40 | 90.85 | 63.23 |
| Qwen2.5-Coder-32B-Instruct | 16.70 | 73.60 | 78.00 | 27.70 | 87.80 | 56.76 |
| HuatuoGPT-o1-72B | 16.70 | 73.00 | 78.00 | 27.40 | 85.37 | 56.09 |
| DeepSeek-R1-Distill-Qwen-32B | 56.70 | 85.60 | 81.00 | 44.70 | 95.73 | 72.75 |
| *Setting for Best of N* | | | | | | |
| Baseline | 66.70 | 90.8 | 75.00 | 44.99 | 96.34 | 74.77 |
| Optimal Setting | 66.70 | 90.8 | 82.20 | 46.20 | 96.34 | 76.45 |
| *Ours v.s. Optimal Setting for Best of N* | | | | | | |
| **CTTS-MM(ours)** | **70.00** | **93.00** | **83.20** | **52.28** | **97.56** | **79.21** |
| *- v.s. Best of N Baseline* | ↑3.30 | ↑2.20 | ↑8.20 | ↑7.29 | ↑1.22 | ↑4.82 |
| *- v.s. Best of N Optimal Setting* | ↑3.30 | ↑2.20 | ↑1.00 | ↑6.08 | ↑1.22 | ↑2.76 |

window. For aggregator, we use Llama-3.3-70B-Instruct. For embedding computation, we adopt Linq-Embed-Mistral Kim et al. (2024) across all experiments, with a fixed embedding dimension of 4,096. For reward models, VLLM is also utilized as inference backend except for Qwen2.5-Math-PRM-7B and Skywork-Reward-V2-Llama-3.1-8B-40M (These two reward models are already fast enough using their huggingface version with Flash Attention). As for other RMs, official VLLM only support Qwen2.5-Math-RM-72B. For other RMs used, we implement their VLLM version by

Table I: PRR accuracy of different reward models on four validation datasets.

| Reward Model | MBPP-Val | MATH-Val | AIME-Val | LiveCodeBench-Val | Avg |
|---|---|---|---|---|---|
| Skywork-Reward-Gemma-2-27B | 61.83 | 51.37 | 50.53 | 49.19 | 53.23 |
| LDL-Reward-Gemma-2-27B-v0.1 | 61.23 | 47.69 | 43.75 | 47.13 | 49.95 |
| Skywork-Reward-V2-Llama-3.1-8B-40M | 58.44 | 76.75 | 82.08 | 80.72 | 74.50 |
| INF-ORM-Llama3.1-70B | 66.98 | 51.51 | 47.43 | 49.29 | 53.80 |
| Qwen2.5-Math-RM-72B | 68.54 | 87.73 | 89.13 | 86.39 | 82.95 |
| Qwen2.5-Math-PRM-7B | 67.05 | 67.15 | 49.32 | 35.74 | 54.82 |
| AceCodeRM-32B | 75.00 | 78.73 | 75.83 | 88.48 | 79.51 |
| QRM-Gemma-2-27B | 61.98 | 49.64 | 45.46 | 53.08 | 52.54 |

ourself (This will be released along with our code). All reward models are set to bfloat16 while other configurations stick to their original settings.

## D.2 HYPERPARAMETERS

For all experiments, we use the same hyperparameters to ensure fair comparison. Specifically, for greedy search process of our ACS, we set top $k = 2$ to initialize our search subset while the number of aggregating is set to 8. For expanding the reward model pool, we consider combinations involving 2 and 3 reward models under three weight method: softmax, linear and sum. The selection number k is set to 100 while the tolerance threshold coefficient $\gamma = 0.95$.

## D.3 DETAILS ON RELATED METHODS

Besides comparing the performance of single LLMs, we also compare our CTTS-MM with five popular multi-LLMs collaboration methods, and the experimental settings are as follows: Symbolic-MoE Chen et al. (2025) retains its original model profiling and LLM selection framework while employing Llama-3.3-70B-Instruct for final response aggregation. MoA Wang et al. (2025) employs 15 LLMs as references, also utilizing Llama-3.3-70B-Instruct as the aggregator. For Self Consistency Chen et al. (2024e), we select the best LLM on the validation datasets of each benchmark to generate eight responses per query, respectively. Majority Voting Chen et al. (2024c) determines the final output through voting among 15 reference LLMs. For Best of N* Snell et al. (2025), N is set to 8. And we use GLM-Z1-32B-0414 as our base model which obtains the highest average accuracy in open-source LLMs while Qwen2.5-Math-RM-72B is utilized as reward model for choosing the best answer as it achieves the best average PRR accuracy as shown in Table I. Like Self Consistency, we also conduct experiments of stronger settings for Best of N, which is shown in Section F.

## D.4 DETAILS ON THE EXPERIMENTS OF INFERENCE TIME

In the manuscript, we investigate the inference time of our CTTS-MM in comparison to other related methods. Here, we provide more details on the experiment setting. Specifically, we report the average inference time per response on MATH-500 and LiveCodeBench with 10 reference LLMs, each deployed on a server with eight A800 GPUs. Although compared with a single LLM, multi-agent systems basically require more LLM forward passes and need more computational resources, most of these forward passes, e.g., the inferences of different referencers and aggregating multiple times, are independent and can be parallelized, making the overall inference time primarily determined by the slowest LLM. For a fair comparison, we apply parallel inference acceleration for all related methods.

## E RESULTS ON COST EFFICIENCY

In the manuscript, we discuss about our analysis on the cost efficiency of out method. Here, we provide qualitative results. As shown in Figure E, we observe that different variants of our CTTS-MM lie along this frontier, in contrast to proprietary LLMs such as GPT-4.1 and Claude Sonnet-3.5, as well as other multi-agent approaches like MOA Wang et al. (2025) and Symbolic-MoE Chen et al. (2025), which are not cost-optimal and incur higher expenses for comparable performance. When quality is the primary objective, CTTS-MM-L represents the best configuration. For scenarios

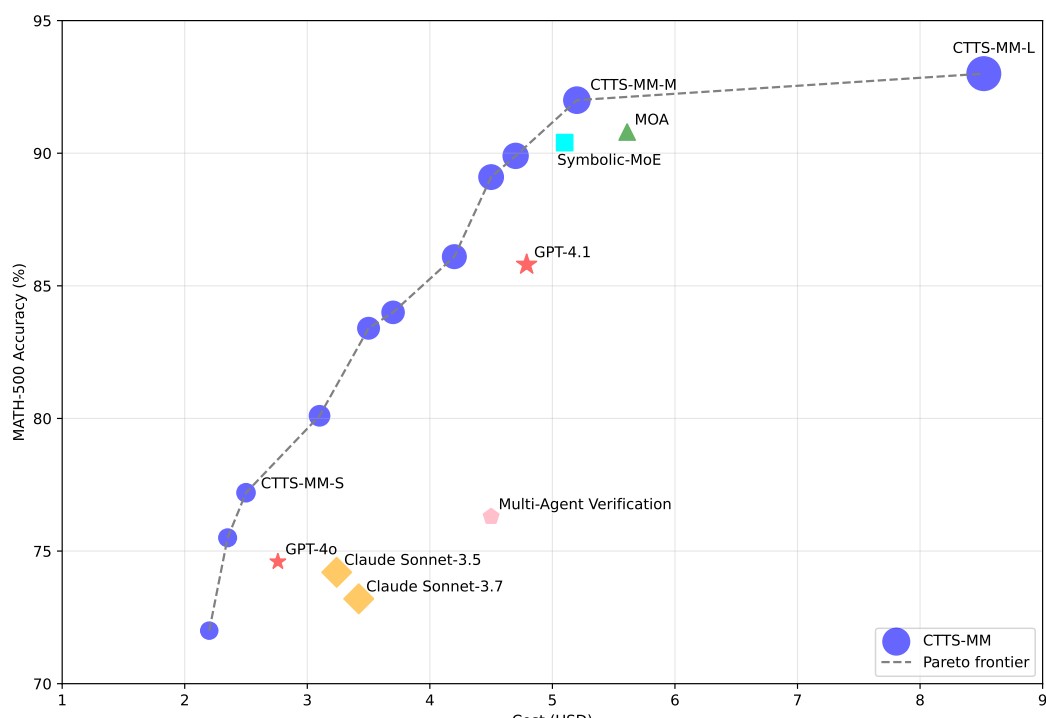

Figure E: Results of performance versus cost on MATH-500.

requiring a favorable trade-off between quality and cost, CTTS-MM-M achieves competitive cost with multi-agent based methods like MOA and Symbolic-MoE or GPT-4.1 while obtains higher level of quality. Notably, it outperforms GPT-4.1 by approximately 6% and Symbolic-MoE by around 2%.

## F  MORE EXPERIMENT RESULTS

### F.1  COMPARISON RESULTS ON TTS PARADIGMS

In our manuscript, we conduct exploratory experiments measuring performance variation among different paradigms. Here, we present additional results on more base models and weight method in Table G. Results basically reveal the same conclusion on Llama-3.3-70B-Instruct and DeepSeek-R1-Distill-Llama-70B as CTTS paradigms outperform both STTS and single- model baselines under most settings.

### F.2  COMPARISON RESULTS ON FIVE BENCHMARKS

We conduct additional experiments to compare our CTTS-MM with Best of N under its optimal setting on AIME, MATH-500, MBPP, LiveCodeBench and Human-eval. The results are shown in Table H. For the baseline of Best of N, we keep it the same with our manuscript, where we use GLM-Z1-32B-0414 as our base model which obtains the highest average accuracy in open-source LLMs while Qwen2.5-Math-RM-72B is utilized as reward model for choosing the best answer. As for Optimal Setting, we select the best open-source LLM on the validation datasets of each benchmark while the best reward model is utilized based on results from Table I. Note that we have no validation split on Human-eval, thus AceCodeRM-32B is utilized since it has best performance on coding benchmark. Results consistently show that our CTTS-MM superior performance increase, still outperforming the optimal setting of Best of N across all five benchmarks and by +2.76% on average accuracy.

Table J: Comparison results of cross-domain question pools.

| Question Pool | AIME | MBPP | MATH-500 | LiveCodeBench |
|---|---|---|---|---|
| MATH-Val | 66.67 | 80.1 | 92.2 | 48.94 |
| MBPP-Val | 66.67 | 82.8 | 91.8 | 50.15 |
| All (Seven Datasets) | 70.00 | 83.2 | 93.00 | 52.28 |

## G  CASE STUDY

We further study the detailed case from our experiments. The detailed contents are provided in Figure F, G and H. We observe that our CTTS-MM is capable of deriving the correct answer through greedy search over the LLM pool using reward scores provided by MoR, even when only one or two models initially produce the correct answer. This result highlights the efficacy of our proposed search process guided by reward scores and underscores the crucial role of our MoR method in providing precise rewards.

## H  ANALYSIS ON QUESTION POOL

We further investigate the impact of utilizing question pools out of domains on overall performance. As shown in Table J, the performance degradation caused by employing out-of-domain question pools in MoR remains marginal. For instance, using an out-of-domain dataset (MBPP) as the question pool for evaluation on a math-related dataset (MATH) results in a marginal performance decrease by 0.4% compared to using an in-domain dataset as the question pool. Similar trends can be observed among other datasets, demonstrating the robustness and stability of our MoR approach. Moreover, when comparing against using a combined question pool from all datasets, we observe consistent performance improvements, highlighting the strong scalability of the question pool.

## I  MORE DETAILS ON ACS

Algorithm A shows the detailed algorithm of our Agent Collaboration Search. Our ACS employs a reward-guided greedy search with early stopping and residual aggregation. Given a set of $n$ candidate answers, ACS first ranks them using a Mixture of Reward Models (MoR) and selects the top-$k$ answers to initialize the search set. An aggregator then combines these into a composite answer, whose reward score is evaluated. Iteratively, ACS examines whether adding remaining candidates improves the aggregated answer. The search terminates when no further improvement is found. Finally, a residual aggregation step merges the best answer with all initial candidates to mitigate information loss, and the output with the higher reward is selected.

## J  DETAILS ON PROMPT

To obtain the optimal task-specific performance across heterogeneous benchmarks, we design prompt individually for each of the seven benchmarks, taking into account their unique characteristics, as shown in Figure I Moreover, the design of the aggregator prompt within our CTTS-MM framework is refined based on MOA Wang et al. (2025), as illustrated in Figure J.

---

**Algorithm A** Greedy Search Paradigm of ACS

---

**Require:** Question $q$, LLM set $D_A$, An initial Answer Set $A_0$, Reward Model set $D_R$, Question Pool $Q_p$, MOR Selective Fuction $F$, Search Steps $T$, Aggregator $Agg$, Initial Search Set Num $k$.

**Ensure:** The optimal answer $A$ to question $q$.

1: **for** $M$ in $D_A$ **do**
2:    $A_0.add(M(q))$                                                                        {**Initialize the answer set.**}
3: **end for**
4: $MOR \leftarrow F(q, Q_p, D_R)$           {**Select suitable reward models or their combinations.**}
5: $Score_0 \leftarrow MOR(A_0)$
6: **for** $i = 1$ to $T$ **do**
7:   **if** $i == 0$ **then**
8:     $chosen\_index \leftarrow Score_0.topk(k).index$
9:     $best\_ans\_set \leftarrow A_0[chosen\_index]$    {**Top k answers ranked by their reward scores constitute the initial search subset.**}
10:    $current\_best \leftarrow Agg(best\_ans\_set)$         {**Aggregate the above top k answers.**}
11:    $current\_best\_score \leftarrow MOR(current\_best)$
12:   **else**
13:     **if** $(A_0 - best\_ans\_set).empty()$ **then**
14:       $break$
15:     **end if**
16:     $improvement \leftarrow False$
17:     $ans\_to\_be\_searched \leftarrow A_0 - best\_ans\_set$
18:     **for** $ans$ in $ans\_to\_be\_searched$ **do**
19:       $temp\_candidate \leftarrow best\_ans\_set$      {**Search for answers with improvement.**}
20:       $temp\_candidate.add(ans)$
21:       $temp\_ans \leftarrow Agg(temp\_candidate)$
22:       $temp\_ans\_score \leftarrow MOR(temp\_ans)$
23:       **if** $temp\_ans\_score \succ current\_best\_score$ **then**
24:         $best\_ans\_set \leftarrow temp\_candidate$
25:         $current\_best \leftarrow temp\_ans$
26:         $current\_best\_score \leftarrow temp\_ans\_score$
27:         $improvement \leftarrow True$
28:       **end if**
29:     **end for**
30:     **if** $!improvement$ **then**
31:       $break$                                      {**If no improvement, stop.**}
32:     **end if**
33:   **end if**
34: **end for**
35: **if** $residual$ **then**
36:   $residual\_set \leftarrow \mathcal{A}, \cup \{current\_best\}$              {**Residual Aggregation**}
37:   $residual\_answer \leftarrow Agg(residual\_set)$
38:   $residual\_score \leftarrow MoR(residual\_answe)$
39:   **if** $residual\_score \succ current\_best\_score$ **then**
40:     $current\_best \leftarrow residual\_answer$
41:   **end if**
42: **end if**
43: $A \leftarrow current\_best$

---

**Case 1 (AIME)**

**Question:** Let $ABC$ be a triangle inscribed in circle $\omega$. Let the tangents to $\omega$ at $B$ and $C$ intersect at point $D$, and let $\overline{AD}$ intersect $\omega$ at $P$. If $AB=5$, $BC=9$, and $AC=10$, $AP$ can be written as the form $\frac{m}{n}$, where $m$ and $n$ are relatively prime integers. Find $m + n$.

**Ground Truth:** 113

**Some Initial Answers from LLM Pool:**

**"Qwen2.5-32b-Instruct":** "To solve for \\(AP\\) in the given triangle \\(ABC\\) inscribed in circle \\(\\omega\\), we start by noting that \\(D\\) is the intersection of the tangents to \\(\\omega\\) at points \\(B\\) and \\(C\\). This implies that \\(DB = DC\\) and that \\(\\angle DBA = \\angle DCA = 90^\\circ - \\angle BAC\\).\n\nSince \\(AD\\) is the symmedian of \\(\\triangle ABC\\), it follows that \\(P\\) is the point where the symmedian intersects the circumcircle again. The length \\(AP\\) can be determined using the formula for the length of a symmedian segment, which is given by:\n\n\\[\nAP = \\frac{AB^2 + AC^2}{BC}\n\\]\n\nGiven the side lengths \\(AB = 5\\), \\(BC = 9\\), and \\(AC = 10\\), we substitute these values into the formula:\n\n\\[\nAP = \\frac{5^2 + 10^2}{9} = \\frac{25 + 100}{9} = \\frac{125}{9}\n\\]\n\nThus, the length \\(AP\\) is \\(\\frac{125}{9}\\). Since 125 and 9 are relatively prime, the fraction is in its simplest form. Therefore, \\(m = 125\\) and \\(n = 9\\), and **the sum \\(m + n\\) is:\n\n\\[\n\\boxed{134}\n\\]**"

**"Meta-Llama-3.3-70B-Instruct":** "**(......We omit some of the thinking steps to save space)** Taking the square root of both sides yields $AP = \sqrt{50} = \frac{\sqrt{50}}{1} = \frac{5\sqrt{2}}{1}$. However, since the answer needs to be in the form $\frac{m}{n}$ where $m$ and $n$ are relatively prime integers, and considering the specific request for the format of the final answer, we need to ensure our solution aligns with the problem's requirements.\n\n## Step 11: Simplify AP to Fit the Required Format\nGiven $AP = \frac{5\sqrt{2}}{1}$, to express it as a fraction with integers and to fit the format required for the solution, we recognize that the value of $AP$ itself doesn't need simplification in terms of removing the radical, but we must remember the question asks for $m + n$ after expressing $AP$ in a certain form. Thus, $AP = \frac{5\sqrt{2}}{1}$ implies $m = 5\sqrt{2}$ and $n = 1$, but since $m$ and $n$ should be integers and relatively prime, the actual values we consider for $m$ and $n$ should directly relate to the fraction form requested, taking into account the simplification and the nature of the numbers involved.\n\n**The final answer is: $\boxed{13}$**"

**"TeleChat2-35B-32K":** "**(......We omit some of the thinking steps to save space)** Using Symmedian Ratio:**\n Using the ratio \\(\\frac{PD}{AD} = \\frac{r}{R}\\), we get:\n \\[ \\frac{PD}{AD} = \\frac{\\sqrt{14}}{2} \\frac{9}{2} = \\frac{\\sqrt{14}}{9}\n \\]\n Let \\( PD = x \\) and \\( AD = y \\). Then:\n \\[\n \\frac{x}{y} = \\frac{\\sqrt{14}}{9} \\quad \\Rightarrow \\quad x = \\frac{\\sqrt{14}}{9} y\n \\]\n From the power of a point:\n \\[\n AP \\cdot AD = 25 \\quad \\Rightarrow \\quad AP = \\frac{25}{AD}\n \\]\n8. **Calculate \\( AD \\) and \\( AP \\):**\n Substitute back into \\( x = \\frac{\\sqrt{14}}{9} y \\)) to find \\( y \\)) and subsequently \\( AP \\)):\n \\[\n y^2 - \\frac{25\\sqrt{14}}{9} y + 25 = 0\n \\]\n Solving this quadratic equation gives \\( y = \\frac{25}{3} \\),\n hence:\n \\[\n AP = \\frac{25}{AD} = \\frac{25}{\\frac{25}{3}} = 3\n \\]\n\nThus, \\( AP = \\frac{3}{1} \\), so \\( m = 3 \\) and \\( n = 1 \\). **Finally, \\( m+n = 3+1 = \\boxed{4} \\).**"

**"GLM-Z1-32B-0414":** "Okay, so I need to find the length of AP in triangle ABC inscribed in a circle \u03c9. **(......We omit some of the thinking steps to save space)** The sides are a=BC=9, b=AC=10, c=AB=5. The semi-perimeter s = (5+9+10)/2 = 24/2 = 12. Then \u0394 = \u221a[s(s-a)(s-b)(s-c)] = \u221a[12(12-9)(12-10)(12-5)] = \u221a[12*3*2*7] = \u221a[504] = \u221a[36*14] = 6\u221a14.\n\nTherefore, the circumradius R = (a b c)/(4 \u0394) = (9*10*5)/(4*6\u221a14) = (450)/(24\u221a14) = (225)/(12\u221a14) = (75)/(4\u221a14). Rationalizing the denominator: 75\u221a14 / (4*14) = 75\u221a14 / 56.\n\nBut maybe I don't need the actual value of R. Let's see.\n\nAlternatively, since D is the intersection of the tangents at B and C **Calculate distance \\(AP\\)**:\n - Coordinates of \\(A\\) and \\(P\\):\n - \\(A = \\left(\\frac{1}{3}, \\frac{4\\sqrt{14}}{3}\\right)\\)\n - \\(P = \\left(\\frac{423}{169}, -\\frac{108\\sqrt{14}}{169}\\right)\\)\n - Distance \\(AP = \\sqrt{\\left(\\frac{1100}{507}\\right)^2 + \\left(\\frac{-1000\\sqrt{14}}{507}\\right)^2} = \\frac{100}{13}\\)\n\nThus, \\(AP = \\frac{100}{13}\\), and \\(**m + n = 100 + 13 = 113**\\).\n\n\\(\\boxed{113}\\)",

**"gemma_3_27b_it":** **(......We omit some of the intermediate answers to save space) The final answer is $\boxed{5486}$**

**"Qwen3-32B":** ". **(......We omit some of the intermediate answers to save space)** let me check calculations again to make sure.\n\nAP squared was calculated as 15,210,000 over 507\u00b2. Then sqrt(15,210,000)=3900 and 507 remains. Then 3900 / 507: divide numerator and denominator by GCD(3900,507). Let me compute GCD(507, 3900). 507 divides into 3900 how many times? 507*7=3549, 3900-3549=351. Now GCD(507, 351). 507=351+156; GCD(351,156). 351=156*2 + 39; GCD(156,39). 156=39*4 + 0. So GCD is 39. Therefore GCD(3900,507)=GCD(507,3900)=39. Therefore divide numerator and denominator by 39:\n\n3900 / 39 = 100, 507 /39 =13. Hence 100/13. Correct. Therefore AP=100/13. So m + n=100 +13=113.\n\n**Therefore the answer is \\boxed{113}**"

**......(answers from other LLM)**

**Search process of CTTS-MM:**

**Win response from Search Step-1 (final reward score: 4.12):** "## Step 1: Understand the given problem and recall the concept of power of a point.\nThe problem involves a triangle $ABC$ inscribed in circle $\omega$, with tangents at $B$ and $C$ intersecting at $D$, and $\overline{AD}$ intersecting $\omega$ again at $P$. We are given the side lengths $AB = 5$, $BC = 9$, and $AC = 10$. To find the length of $AP$, we need to apply the concept of power of a point.\n\n## Step 2: Apply the power of a point theorem to find $DB^2$.\nBy the power of a point theorem, $DB^2 = DP \cdot DA$. Let's denote $AP$ as $x$ and $BP$ as $y$. Then, $DP = x + 5$ and $DA = 5 + x$. Substituting these expressions into the equation $DB^2 = DP \cdot DA$, we get $DB^2 = (x + 5)(5 + x)$.\n\n## Step 3: Use the given information about the triangle $ABC$ to determine $DB$.\nUsing the Law of Cosines on \\(\triangle ABC\\), we can find $\\cos A$. With $\\cos A$ known, we can further calculate $\\sin A$. Then, applying the Law of Sines to \\(\triangle ABD\\), we can relate $AD$ and $DB$. With $AD$ known, we can then solve for $DB$.\n\n## Step 4: Solve for $AP$ by equating the two expressions for $DB^2$.\nAfter obtaining $DB$ from the previous step, we equate it to $(x + 5)(5 + x)$ and solve for $x$, which represents $AP$.\n\n## Step 5: Express $AP$ in the form $\frac{m}{n}$ where $m$ and $n$ are relatively prime integers and find $m + n$.\nOnce $AP$ is found, express it in the required form $\frac{m}{n}$, ensuring that $m$ and $n$ are relatively prime. Finally, compute $m + n$.\n\n**The final answer is: $\boxed{13}$**"

**Win response from Search Step-2 (final reward score: 10.36):** "## Step 1: Apply the Power of a Point Theorem\nTo solve this problem, we will use the Power of a Point theorem. This theorem states that for any point $P$ outside a circle with points of intersection $X$ and $Y$ with the circle, the product of the lengths $PX$ and $PY$ is constant for any line through $P$ to the circle. In our case, we consider point $D$ as $P$ and apply the theorem to find a relationship between the lengths of segments intersecting the circle.\n\n## Step 2: Find the relationship using Power of a Point\nBy Power of a Point from $D$, we have $DB \cdot DC = DP \cdot DA$. Since $DB$ and $DC$ are tangents from the same external point to the circle, and given $AB = 5$, $BC = 9$, $AC = 10$, we need to leverage these to find expressions for $DB$ and $DC$.\n\n## Step 3: Calculate DB and DC\nUsing the fact that in a triangle, the lengths of the tangents from an external point to the circle are equal, let's denote $x = DB$ and $y = DC$. We also know that $DA = DP + PA$. However, to directly apply Power of a Point, we recognize the need to express $DB$ and $DC$ in terms of $AB$, $BC$, and $AC$, but directly calculating them requires additional relationships.\n\n## **(......We omit some of the thinking steps to save space)**
Step 11: Simplify AP to Fit the Required Format\nGiven $AP = \frac{5\sqrt{2}}{1}$, to express it as a fraction with integers and to fit the format required for the solution, we recognize that the value of $AP$ itself doesn't need simplification in terms of removing the radical, but we must remember the question asks for $m + n$ after expressing $AP$ in a certain form. Thus, $AP = \frac{5\sqrt{2}}{1}$ implies $m = 5\sqrt{2}$ and $n = 1$, but since $m$ and $n$ should be integers and relatively prime, the actual values we consider for $m$ and $n$ should directly relate to the fraction form requested, taking into account the simplification and the nature of the numbers involved.\n\n**The final answer is: $\boxed{113}$**".

**......(omit intermediate search steps (step-3 final score: 12.37, step-4 final score: 14.09))**

**Win response from Search Step-5 (final reward score: 16.25):** "## Step 1: Apply the Power of a Point theorem\nThe Power of a Point theorem states that for any point $D$ outside a circle with tangents drawn and intersecting chords as well, the product of the lengths of one tangent segment and its external secant segment is equal to the square of the length of the tangent segment from $D$ to the point of tangency. In this problem, we're given $AB = 5$, $BC = 9$, and $AC = 10$. The theorem tells us $DB^2 = DP \cdot DA$.\n\n## Step 2: Calculate $DB$\nBecause $\triangle ABC$ is a right triangle (given $AB = 5$, $BC = 9$, $AC = 10$, which does not satisfy Pythagoras' theorem for an integer-sided right triangle but let's proceed with calculations), we can find the circumradius using the formula $R = \frac{abc}{4\Delta}$ where $\Delta$ is the area of $\triangle ABC$. However, recognizing the Pythagorean triple $(5, 12, 13)$ and comparing with our sides, our triangle isn't a Pythagorean triple. Instead, we should directly apply the Power of a Point theorem after finding the necessary lengths.\n\n## Step 3: Identify relevant relationships for $DA$ and $DP$\nSince $AP = DP - DA$, we need to express $DA$ and $DP$ in terms of known quantities or find a direct way to calculate $AP$. To do so, let's utilize the properties of inscribed angles, central angles, and the fact that $D$ is the intersection of tangents.\n\n## Step 4: Calculate $DA$ or $DP$\nTo proceed further, let's recall the relation $DB^2 = DA \cdot DP$. Without loss of generality, assume we have $DB$ and need $DA$ or $DP$. Knowing $AB$, $BC$, and $AC$, and that $D$ is external to $\omega$, we might relate these lengths through similar triangles or trigonometric ratios, considering the angles formed at $D$ and within $\triangle ABC$.\n\n## Step 5: Consider alternative approaches to find $AP$\nGiven the complexities in calculating $DB$ and $DA$ or $DP$ directly from previous steps without explicit angle measures or side lengths involving $D$, consider leveraging geometric properties such as similar triangles, the intersecting chord theorem, or utilizing the Law of Cosines and Law of Sines to derive relationships between the segments.\n\n## Step 6: Reflect on provided solutions for insight\nConsidering the complexity of direct calculation without specific trigonometric values or additional geometric constructions, reflect on the approach provided in the rationale which calculates $AP$ using the identified relationship $AP = \frac{100}{13}$ by leveraging geometric principles and the Power of a Point theorem correctly applied to the given configuration.\n\n**The final answer is: $\boxed{113}$**"

**Win response from Search Step-6 (final reward score: 11.43):...**

**The reward score decrease, Stop Here. The final answer will be the one from Step-5**

Figure F: Detailed case results from AIME.

**Case 2 (GPQA)**

**Question:** You have prepared a di-substituted 6-membered aromatic ring compound. The FTIR spectrum of this compound shows absorption peaks indicating the presence of an ester group. The 1H NMR spectrum shows six signals: two signals corresponding to aromatic-H, two signals corresponding to vinyl-H (one doublet and one doublet of quartets), and two signals corresponding to –CH3 groups. There are no signals corresponding to –CH2 groups. Identify the chemical formula of this unknown compound as either C11H12O2, C11H14O2, C12H12O2, or C12H14O2.

**Option:** ["C11H14O2", "C12H12O2", "C11H12O2", "C12H14O2"]

**Ground Truth: B**

**Some Initial Answers from LLM Pool:**

**"Qwen2.5-32b-Instruct":** "Let's think step by step:\n\n1. **Aromatic Ring and Substitution**:\n - A 6-membered aromatic ring (benzene) has 6 carbon atoms and 6 hydrogen atoms.\n - The compound is disubstituted, meaning there are two substituents on the benzene ring.\n\n2. **Ester Group**:\n - The presence of an ester group (C=O and -O-) is confirmed by the FTIR spectrum.\n - An ester group contains 2 carbon atoms and 4 oxygen atoms (considering the rest of the molecule).\n\n3. **NMR Analysis**:\n - The 1H NMR spectrum indicates:\n - Two signals for aromatic-H: This suggests the aromatic protons are split into two groups, possibly due to different substitution patterns.\n - Two signals for vinyl-H (one doublet and one doublet of quartets): This indicates the presence of a vinyl group (-CH=CH-), contributing 2 more carbon atoms and 2 hydrogen atoms.\n - Two signals for -CH3 groups: Two methyl groups contribute 2 carbon atoms and 6 hydrogen atoms.\n\n4. **No -CH2 Groups**:\n - The absence of -CH2 groups means no methylene groups are present in the molecule.\n\n5. **Calculating the Formula**:\n - Benzene (C6H6) + 2 substituents:\n - Considering the aromatic and vinyl-H signals, the substituents are likely a vinyl group (-CH=CH2) and a -CH3 group.\n - Vinyl group: 2 C, 3 H.\n - Methyl group: 1 C, 3 H.\n - Total from benzene and substituents: C6 + 2C + 1C = 9C, H6 + 2H + 3H = 11H.\n - Adding the ester group (C=O and -O-): 1C + 2O but O is already counted as part of the functional group.\n - Total: C9 + 1C (ester) = C10, H11.\n - To balance the formula, we need to consider the -CH3 group (which we counted earlier) and the ester group together, giving us a total of 2 more carbons (C12) and 2 more hydrogens (H14).\n\n**Therefore, the correct chemical formula is C12H14O2.\n\nThe correct answer is (D).**"

**"Meta-Llama-3.3-70B-Instruct":** "To determine the chemical formula of the unknown compound, let's analyze the information provided by the FTIR and 1H NMR spectra.\n\n1. **FTIR Spectrum**: The presence of an ester group is indicated by specific absorption peaks. Ester groups are characterized by strong absorptions around 1735-1800 cm^-1 for the C=O stretch and 1000-1300 cm^-1 for the C-O stretch. This information confirms the presence of an ester functionality in the compound but does not directly inform us about the overall molecular formula.\n\n2. **1H NMR Spectrum**:\n - **Aromatic-H**: Two signals indicate that the aromatic ring is di-substituted. Given that it's a 6-membered aromatic ring (like benzene), a di-substitution pattern would result in two distinct signals for the aromatic hydrogens if the substituents are different, assuming the substituents do not cause the ring to the information:\n- The compound has an ester group.\n- It has a di-substituted aromatic ring.\n- There's a vinyl group with an adjacent methyl group.\n- There are no –CH3 groups, which means the methyl groups are directly attache become symmetrical (which would reduce the number of distinct signals).\n - **Vinyl-H**: The presence of one doublet and one doublet of quartets suggests there's an alkene (vinyl group) with one hydrogen being coupled to another hydrogen (doublet) and the other being coupled to both another hydrogen and a methyl group (doublet of quartets), indicating an allylic methyl group.\n - **No –CH3 Groups**: Two signals for methyl groups suggest there are at least two methyl groups in the compound. One is likely part of the ester group (if it's a methyl ester), and the other could be the allylic methyl mentioned above.\n\nGiven that any methylene groups.\n - **C11H14O2**: This adds two more hydrogens than C11H12O2, which could better accommodate the methyl groups and the vinyl hydrogens, considering the aromatic hydrogens are already accounted for.\n- **C12H12O2**: This formula adds a carbon but no additional hydrogens compared to C11H12O2, which doesn't fit well with the need for additional hydrogens to account for the methyl and vinyl groups.\n- **C12H14O2**: This adds both a carbon and two hydrogens compared to C11H12O2, offering a good balance for the aromatic ring, the ester group, the vinyl group, and the two methyl groups.\n\nConsidering the need to account for the di-substituted aromatic ring, the ester group, the vinyl group with its adjacent methyl, and another methyl group (likely as part of the ester), **C12H14O2 seems to provide the most plausible balance of carbons and hydrogens to accommodate all these functionalities without requiring –CH3 groups.\n\nThe correct answer is (D)"**

**"TeleChat2-35B-32K":** "(......We omit some of the thinking steps to save space) Another possibility is a phenyl group attached to the benzene ring, with an ester group and a vinyl group with two methyl groups.\n\nGiven the number of hydrogen atoms in the aromatic and vinyl groups, and considering the absence of methylene groups, the simplest structure fitting the description is:\n\n- A benzene ring (C6H6)\n- An ester group (C2086H\u2084)\n- An ester group (\u2013COOCH\u2083, contributing 3 carbons, 1 oxygen, and 3 hydrogens)\n- A vinyl group with two methyl groups (\u2013CH=CH\u2083CH\u2083, contributing 4 carbons and 3 hydrogens)\n\nAdding these up:\n- Total carbons: \\(6 + 3 + 4 = 13\\)\n- Total hydrogens: \\(4 + 3 + 3 = 10\\)\n- Total oxygens: \\(1\\)\n\nSo, the molecular formula should be C\u2081\u2083H\u2081\u2080O\u2082.\n\nBut looking at the choices given, the closest match is C\u2081\u2082H\u2081\u2082O\u2082, which would correspond to a slightly different arrangement but still fitting the criteria of having an aromatic ring, an ester group, and a vinyl group with two methyl groups.\n\n**Therefore, the correct answer is (B)."**

**"GLM-Z1-32B-0414":** "Okay, let's try to figure this out. So the question is about identifying a disubstituted aromatic compound based on FTIR and NMR data. The possible options are C11H12O2, C11H14O2, C12H12O2, or C12H14O2.\n\nFirst, the FTIR shows an ester group. Esters have characteristic absorption peaks around 1740 cm\u207b\u00b9 for the carbonyl (C=O) stretch. So the compound has an ester functional group, which contributes an oxygen atom. The general formula for an ester is RCOOR', so that's two oxygen atoms.\n\nWait, but the molecular formulas given all have O2, so that checks out. So each of these compounds has two oxygen atoms, which fits with being an ester.\n\nNext, the 1H NMR. There are six signals: two aromatic-H signals, two vinyl-H (one doublet and one doublet of quartets), and two -CH3 groups. No -CH2 groups. Let's break this down.\n\nThe aromatic protons are two signals. Since it's a di-substituted benzene, the substitution pattern affects the splitting. For example, para-substituted would have two equivalent protons each (if symmetric), but maybe ortho or meta. But the exact pattern isn't clear yet. The presence of two signals suggests maybe two different sets of aromatic protons, like in a para-disubstituted ring with different substituents, or maybe something else.\n\nThe vinyl protons: one doublet and one doublet of quartets. That sounds like a vinyl group adjacent to another group. A doublet of quartets usually comes from a proton that's next to two different groups. For example, in a structure like CH2CH2 where there's coupling between adjacent protons, but wait, the description says no -CH2 groups. Cinnamate is benzene with a COOCH3 group and a CH=CH2 group. **But that would have a CH2. \n\nI'm stuck. Let's look at the answer choices again. The correct answer is likely B)** C12H12O2 or C) C11H12O2.\n\n**(......We omit some of the thinking steps to save space)** Based on the DoU and the need for an additional double bond, C11H12O2 has DoU 6 (benzene 4 + ester 1 + 1 double bond). C12H12O2 has DoU 7, which would require an extra DoU. \n\nIf the structure has a benzene, ester, and a double bond, then C11H12O2 is possible. For example, the ester is COOCH3 and the other substituent is CH=CH2. Let's count the carbons: C6 (benzene) + 2 (COO) + 2 (CH=CH2) =10. Need one more carbon. Maybe the other substituent is C2H3O?\n\nNot sure. Alternatively, the other substituent is a propenyl group (CH2=CHCH2-), but that has CH2. \n\nAlternatively, the other substituent is a vinyl group (CH=CH-) attached to the benzene and connected to a methyl. For example, CH2=CHCH3, but that's propene, which has a CH2. \n\nWait, maybe the other substituent is a cyclopropyl group with a methyl."

**"gemma_3_27b_it":** "Let's break down the information provided to deduce the chemical formula.\n\n1. **6-membered aromatic ring:** This means a benzene ring (C6H6) is present as the core structure.\n\n2. **Di-substituted:** Two substituents are attached to the benzene ring.\n\n3. **(......We omit some of the thinking steps to save space)** . Plus 2 O from the ester group gives us C10H14O2.\n\nBut the question states that the compound is di-substituted. That means that the vinyl group has to have another carbon, making it a propenyl group. If we assume the structure is a benzene ring with a vinyl group (CH=CH2) and a methyl ester (COOCH3), we get:\n\nC6 (benzene) + C3 (propenyl) + C2 (ester) + O2 (ester) = C11 and H6 + H6 + H4 + H2 = H18\n\nThis doesn't match any of the choices, so it seems we need to rethink the vinyl group.\n\nLet's think about an ether, and a vinyl group. This will be:\n\nC6 + C4 + C2 + O2 = C14 atoms\n\nnH6 + H8 + H4 + H2 = H20 atoms.\n\nThis is not correct.\n\nHowever, the compound is di-substituted. Consider a vinyl ester group.\n\nThe ester structure could be CH3-CO-O-CH=CH2. This will give us C6 (benzene ring) + C2 (ester) + C3 (vinyl) + O2 (ester) = C13, and H6 + H4 + H4 + H2 = H16. This is not correct.\n\nIt looks like the vinyl must be connected to the ester.\n\nThe correct answer should be C11H12O2. This is because the correct molecular formula for the ester is C2H4O2 and the vinyl group is C9H4.\n\nAdding this to benzene (C6H6) yields C11H12O2.\n\n**The correct answer is (C)"**

**"Qwen3-32B":** "<think>\nOkay, let's try to figure this out. So the question is about identifying the chemical formula of a di-substituted 6-membered aromatic ring compound based on its FTIR and 1H NMR spectra. The possible options are C11H12O2, C11H14 **(......We omit some of the thinking steps to save space)** No CH2 groups are detected. \n\n2. **Aromatic Substitution Pattern:** \n - Two aromatic signals suggest a **symmetrical disubstitution** (e.g., para-substitution). This minimizes the number of distinct aromatic proton environments. \n\n3. **Substituent Analysis:** \n - The **ester group** (e.g., -COOCH3) contributes 2 carbons (carbonyl and methyl) and one methyl signal. \n - The **vinyl group** (e.g., -CH2=C(CH3)) contributes 3 carbons (propenyl) and another methyl signal. \n\nTogether, these substituents add **5 carbons** to the benzene ring (6 + 5 = 11 total carbons). \n\n4. **Hydrogen Count and Degrees of Unsaturation:** \n - The molecular formula **C11H14O2** corresponds to **5 degrees of unsaturation** [(2\u00d711 + 2 - 14)/2 = 5]. This matches the expected contributions: \n - Benzene ring: 3 (3 double bonds + 1 ring). \n - Ester: 1 (C=O double bond). \n - Vinyl group: 1 (double bond). \n - The absence of CH2 groups and the presence of two methyl groups align with the substituent structures proposed. \n\n5. **Elimination of Other Options:** \n - **(A) C11H12O2** and **(D) C12H14O2** have 6 degrees of unsaturation, which would require an additional double bond or ring, inconsistent with the NMR data. \n - **(C) C12H12O2** has 7 degrees of unsaturation, exceeding the required 5. \n\nThus, **C11H14O2 (B)** is the only formula that satisfies all spectroscopic and structural **clues. \n\n**The correct answer is (B).**",
"

**......(answers from other LLM)**

**Search process of CTTS-MM:**

**Win response from Search Step-1 (final reward score: -2.35):** "To solve this, let's analyze the given information step by step:\n\n1. **FTIR Spectrum**: Shows absorption peaks indicating the presence of an ester group. This means the compound contains at least one ester functional group (-COO-), which contributes to the oxygen count in the molecule.\n\n2. **1H NMR Spectrum**:\n - **Aromatic-H**: Two signals indicate a di-substituted benzene ring, likely suggesting symmetry (like para-substitution).\n - **Vinyl-H**: Two signals (a doublet and a doublet of quartets) imply the presence of a vinyl group (CH=CH2 or similar) with specific coupling patterns.\n - **-CH3 groups**: Two signals with no -CH2 groups suggest the methyl groups are directly attached to other carbons without intervening methylene groups.\n - **No -CH2 signals**: This is crucial as it limits the possible structures, implying that any alkyl chains must be branched or part of a ring/vinyl structure without methylene groups.\n\n3. **Chemical Formula Options**:\n - **C11H12O2**\n - **C11H14O2**\n - **C12H12O2**\n - **C12H14O2**\n\nLet's calculate the degrees of unsaturation (DoU) for each option to narrow down the possibilities:\n- DoU = (2C + 2 - H)/2,\n\nwhere C is the number of carbon atoms, H is the number of hydrogen atoms, X is the number of halogen atoms, and N is the number of nitrogen atoms. Assuming no halogens or nitrogen, the formula simplifies to: DoU = (2C + 2 - H)/2.\n\nFor each option:\n- **C11H12O2**: DoU = (22 + 2 - 12)/2 = 12/2 = 6\n- **C11H14O2**: DoU = (22 + 2 - 14)/2 = 10/2 = 5\n- **C12H12O2**: DoU = (24 + 2 - 12)/2 = 12/2 = 6\n- **C12H14O2**: DoU = (24 + 2 - 14)/2 = 12/2 = 6\n\nThe benzene ring itself has 4 DoU (3 from the double bonds and 1 from the ring). The ester group contributes 1 DoU (from the carbonyl double bond), and the vinyl group contributes another 1 DoU. Thus, the minimum expected DoU from these functionalities is 6 (4 from benzene + 1 from ester + 1 from vinyl).\n\nOptions **C11H12O2** and **C12H14O2** both have a DoU of 6, which matches our expectations on the functionalities mentioned. However, **C12H14O2** fits better with the idea of having additional hydrogens that could come from methyl groups attached to the benzene on the vinyl group.\n\nConsidering the need for two methyl groups and a vinyl group without any -CH2 groups, and matching the DoU, **C12H14O2** seems more plausible as it allows for the additional hydrogens from the methyl groups and fits the DoU calculation.\n\n**The correct answer is (C)"**

**Win response from Search Step-2 (final reward score: 11.97):** "To solve this, let's analyze the given information step by step:\n\n1. **FTIR Spectrum**: The presence of an ester group is indicated by specific absorption peaks. Ester groups have a general formula of RCOOR', where R and R' are hydrocarbon chains. This means the compound contains at least two oxygen atoms.\n\n2. **1H NMR Spectrum**:\n - **Aromatic-H**: Two signals suggest a symmetric disubstitution pattern on the benzene ring (e.g., para-substitution).\n - **Vinyl-H**: A doublet and a doublet of quartets indicate a vinyl group (CH=CH2) with specific coupling patterns, suggesting it's part of a substituent.\n - **-CH3 Groups**: Two signals for methyl groups without any -CH2 signals might these methyl groups are directly attached to other carbons or part of the ester group (like OCH3).\n\n3. **No -CH2 Signals**: This implies that all carbon atoms in the compound are either part of the aromatic ring, a methyl group, a vinyl group, or part of the ester functionality without any methylene (-CH2-) groups.\n\n4. **Chemical Formula Options**:\n - **C11H12O2**\n - **C11H14O2**\n - **C12H12O2**\n - **C12H14O2**\n\nLet's calculate the degrees of unsaturation (DoU) for each option, which can help narrow down the possibilities:\n- DoU = (2C + 2 + N - H - X)/2,\n\nwhere C is the number of carbon atoms, N is the number of nitrogen atoms, H is the number of hydrogen atoms, and X is the number of halogen atoms. Since there are no mentions of nitrogen or halogens, the formula simplifies to DoU = (2C + 2 - H)/2.\n\nFor each option:\n- **C11H12O2**: DoU = (22 + 2 - 12)/2 = 6\n- **C11H14O2**: DoU = (22 + 2 - 14)/2 = 5\n- **C12H12O2**: DoU = (24 + 2 - 12)/2 = 7\n- **C12H14O2**: DoU = (24 + 2 - 14)/2 = 6\n\nThe benzene ring itself counts as one.\n\nAn ester group contributes 1 degree of unsaturation (from the carbonyl double bond), and a vinyl group also contributes 1 degree of unsaturation.\n\nGiven the presence of a vinyl group and an ester, we would expect at least 6 degrees of unsaturation (4 from the benzene + 1 from the ester + 1 from the vinyl group).\n\nThis matches options **C11H12O2** and **C12H14O2**, both of which have 6 degrees of unsaturation.\n\nHowever, considering the lack of -CH2 groups and the specific NMR signals, we need to account for the structure that fits the given data. The absence of -CH2 signals suggests that all hydrogens are either on the aromatic ring, part of a methyl group, or part of the vinyl group.\n\nGiven the complexity and the specific details provided, let's focus on the most critical aspect: the degrees of unsaturation and the NMR data fitting the structure. The correct approach involves matching the NMR and FTIR data with possible structures that fit the molecular formulas provided, considering the degrees of unsaturation and the absence of -CH2 groups.\n\n**The correct answer is (B)"**

**Win response from Search Step-3 (final reward score: 8.46):** "Since there are no mentions of nitrogen or halogens, the formula simplifies to DoU = (2C + 2 - H)/2.\n\nFor each option:\n- **C11H12O2**: DoU = (22 + 2 - 12)/2 = 12/2 = 6\n- **C11H14O2**: DoU = (22 + 2 - 14)/2 = 10/2 = 5\n- **C12H12O2**: DoU = (24 + 2 - 12)/2 = 14/2 = 7\n- **C12H14O2**: DoU = (24 + 2 - 14)/2 = 12/2 = 6\n\nThe benzene ring itself contributes 4 degrees of unsaturation (3 double bonds and the ring itself counts as one). An ester group contributes 1 degree of unsaturation (from the carbonyl double bond), and a vinyl group also contributes 1 degree of unsaturation.\n\nGiven the presence of a vinyl group and an ester, we would expect at least 6 degrees of unsaturation (4 from the benzene + 1 from the ester + 1 from the vinyl group). This matches options **C11H12O2** and **C12H14O2**, both of which have 6 degrees of unsaturation.\n\nHowever, considering the lack of -CH2 groups and the specific details provided, let's focus on the most critical aspect: the degrees of unsaturation and the NMR data fitting the structure. The correct approach involves matching the NMR and FTIR data with possible structures that fit the molecular formulas provided, considering the degrees of unsaturation and the absence of -CH2 groups.\n\n**The correct answer is (B)"**

**The reward score decrease, Stop Here. The final answer will be the one from Step-2.**

Figure G: Detailed case results from GPQA.

1242
1243
1244
1245
1246
1247
1248
1249
1250
1251
1252
1253
1254
1255
1256
1257
1258
1259
1260
1261
1262
1263
1264
1265
1266
1267
1268
1269
1270
1271
1272
1273
1274
1275
1276
1277
1278
1279
1280
1281
1282
1283
1284

**Case 3 (MBPP)**

**Question:** Write a function to get the angle of a complex number. Your code should satisfy the following assertion with the same function name:\n```python\nassert angle_complex(0,1j)==1.5707963267948966 \n```

**Some Initial Answers from LLM Pool:**
**"Qwen2.5-32b-Instruct"**(exec result: **false**): "To calculate the angle of a complex number, you can use the `cmath` library in Python, which provides a function to find the phase (angle) of a complex number. Here is how you can write the function:\n\n```python\nimport cmath\n\ndef angle_complex(real, imag):\n # Create a complex number from real and imaginary parts\n complex_num = complex(real, imag)\n # Calculate the angle (phase) of the complex number\n angle = cmath.phase(complex_num)\n return angle\n\n# Test the function with the provided assertion\nassert angle_complex(0, 1j) == 1.5707963267948966\n```\n\nIn this code:\n- `complex(real, imag)` creates a complex number from the real and imaginary parts.\n- `cmath.phase(complex_num)` calculates the phase (angle) of the complex number in radians."

**"Meta-Llama-3.3-70B-Instruct"** (exec result: **false**): "Of course! I'll guide you through creating a function to calculate the angle of a complex number, ensuring it meets the provided assertion. Here is the Python implementation:\n\n```python\nimport math\ndef angle_complex(real, imag):\n return math.atan2(imag, real)\n```\n\n### Explanation\n\n* The function `angle_complex` takes two parameters, `real` and `imag`, representing the real and imaginary parts of a complex number, respectively.\n* It utilizes the `math.atan2` function from Python's math library, which returns the arc tangent of `imag` divided by `real` in radians. The `atan2` function is used instead of `atan` to handle the case where `real` is zero and to get the correct quadrant.\n* The `math.atan2` function automatically handles the signs of `real` and `imag` to determine the correct angle in the range of $(-\\pi, \\pi]$.\n* The calculated angle is then returned by the `angle_complex` function, satisfying the given assertion."

**"TeleChat2-35B-32K"** (exec result: **false**): "To get the angle of a complex number in Python, you can use the `cmath` module, which provides functions to work with complex numbers. The `cmath.phase` function returns the phase (angle) of a complex number in radians.\n\nHere is the function that satisfies the given assertion:\n\n```python\nimport cmath\n\ndef angle_complex(complex_num):\n return cmath.phase(complex_num)\n\n# Test the function with the given assertion\nassert angle_complex(0+1j) == 1.5707963267948966\n```\n\nThis function takes a complex number as input and returns its angle in radians. The provided assertion checks that the function correctly computes the angle of the complex number \\(0 + 1j\\), which should be \\(\\frac{\\pi {2}\\) radians or approximately 1.5707963267948966."

**"GLM-Z1-32B-0414"** (exec result: **true**):: "Okay, I need to write a Python function called angle_complex **(......We omit some of the intermediate answers to save space)** The `cmath.phase()` function returns the angle in radians between the positive real axis and the complex number in the complex plane.\n\n### Solution Code\n```python\nimport cmath\n\ndef angle_complex(real, imag):\n z = real + imag\n return cmath.phase(z)\n```\n\n### Explanation\n- **Combining Real and Imaginary Parts**: The function takes two arguments, `real` and `imag`. By adding these two arguments directly, we form the complex number. For example, if `real` is 0 and `imag` is 1j, the result of `real + imag` is the complex number `0 + 1j`.\n- **Calculating the Phase**: The `cmath.phase()` function is used to compute the angle of the complex number. This function handles all edge cases, including when the real part is zero, ensuring the correct angle is returned.\n\nThis approach efficiently constructs the complex number and computes its angle using Python's built-in capabilities, ensuring accuracy and reliability."

**"gemma_3_27b_it"** (exec result: **false**): "```python\nimport cmath\n\ndef angle_complex(x, y):\n \"\"\"\n Calculates the angle of a complex number.\n\n Args:\n x: The real part of the complex number.\n y: The imaginary part of the complex number.\n\n Returns:\n The angle of the complex number in radians.\n \"\"\"\n return cmath.phase(complex(x, y))\n```"

**"Qwen3-32B"** (exec result: **false**): "<think>\nOkay, I need to write a Python function called angle_complex **(......We omit some of the intermediate answers to save space)** Implementation\n\nHere is the implementation of the `angle_complex` function:\n\n```python\nimport cmath\n\ndef angle_complex(a, b):\n \"\"\"\n Returns the angle (in radians) of a complex number constructed\n from the real part of 'a' and the imaginary part of 'b'.\n\n Parameters:\n a (complex or real): Real part of the complex number.\n b (complex or real): Imaginary part of the complex number.\n\n Returns:\n float: The phase (angle) of the resulting complex number.\n \"\"\"\n real = a.real\n imag = b.imag\n return cmath.phase(complex(real, imag))\n```\n\n---\n\n### Explanation\n\n- **`a.real`**: Extracts the real component from the first argument.\n- **`b.imag`**: Extracts the imaginary component (the coefficient of `j`) from the second argument.\n- **`complex(real, imag)`**: Constructs the complex number `real + imag*j`.\n- **`cmath.phase(...)`**: Computes the angle (argument) of the complex number in radians.\n\n---\n\n### Test Case\n\n```python\n>>> angle_complex(0, 1j)\n1.5707963267948966\n```\n\nThis satisfies the provided assertion and works as expected.\n\n---\n\n### Edge Case Handling\n\n> Note: If both `real` and `imag` are zero, `cmath.phase(complex(0,0))` will raise a `ValueError`. It's important to ensure the function is used with valid inputs representing a non-zero complex number."

**……(answers from other LLM)**

**Search process of CTTS-MM:**

**Win response (extracted solution) from Search Step-1 (final reward score: 1.3, exec result: false):** "```python\nimport math\n\ndef angle_complex(real, imag):\n # Check if the input is a complex number\n if isinstance(imag, complex):\n imag = imag.imag\n \n return math.atan2(imag, real)\n```"

**Win response from Search Step-2 (final reward score: 5.27, exec result: false):** "```python\nimport math\n\ndef angle_complex(real, imag):\n z = real + imag\n return math.atan2(z.imag, z.real)\n```"

**Win response from Search Step-3 (final reward score: 16.57, exec result: true):** "```python\nimport cmath\n\ndef angle_complex(real, imag):\n # Combine real and imag into a complex number\n z = real + imag\n # Return the phase (angle) of the complex number\n return cmath.phase(z)\n\n# Test the function with the given assertion\nassert angle_complex(0, 1j) == 1.5707963267948966\n```"

**Win response from Search Step-4 (final reward score: 16.57 , exec result: true):** "```python\nimport cmath\n\ndef angle_complex(real, imag):\n # Combine real and imag into a complex number\n z = real + imag\n # Return the phase (angle) of the complex number\n return cmath.phase(z)\n\n# Test the function with the given assertion\nassert angle_complex(0, 1j) == 1.5707963267948966\n```"

**The reward score ceases increasing, Stop Here. The final answer will be the one from Step-3**

Figure H: Detailed case results from MBPP.

1285
1286
1287
1288
1289
1290
1291
1292
1293
1294
1295

### Prompt for MBPP benchmark

**System Prompt:** "You are an exceptionally intelligent coding assistant that consistently delivers accurate and reliable responses to user instructions."
**User Prompt:** "Question: {question}."

### Prompt for LiveCodeBench benchmark

**System Prompt:** "You are an expert Python programmer. You will be given a question (problem specification) and will generate a correct Python program that matches the specification and passes all tests."
**User Prompt:** "Question: {question}."

### Prompt for Human-eval benchmark

**System Prompt:** "You are an expert Python programmer. You will be given a coding question (problem specification) and will generate a correct Python program that matches the specification and passes all tests. Directly give the executable function body, without any comments or test cases."
**User Prompt:** "Question: {question}."

### Prompt for AIME benchmark

**System Prompt:** "Please reason step by step, and put your final answer within \\boxed{}."
**User Prompt:** "Question: {question}."

### Prompt for MATH benchmark

**System Prompt:** "You are a math problem solver. Please solve the following math problem. Be sure to explain your solution in detail. The numerical values in the answer should be surrounded by \\boxed. The final answer should start with 'The answer is' and give the conclusion directly. Do not add any extra content."
**User Prompt:** "Question: {question}."

### Prompt for GPQA benchmark

**System Prompt:** "You are a very intelligent assistant, who follows instructions directly."
**User Prompt:** "Question: {question}."

### Prompt for IFEval benchmark

**User Prompt:** "Instruction: {question}."

Figure I: Prompts for seven benchmarks.

> **Prompt for Aggregator**
>
> **System Prompt:** "You have been provided with a set of responses from various open-source models to the latest user query. Your task is to synthesize these responses into a single, high-quality response. It is crucial to critically evaluate the information provided in these responses, recognizing that some of it may be biased or incorrect. Your response should not simply replicate the given answers but should offer a refined, accurate, and comprehensive reply to the instruction. Ensure your response is well-structured, coherent, and adheres to the highest standards of accuracy and reliability.
> Responses from models:
> 1.{Response1}
> 2.{Response2}
> ... "
> **User Prompt:** "Question: {question}."

Figure J: Prompt for Aggregator within our CCTS-MM

