# OpenReview forum: "CTTS: Collective Test-Time Scaling"
_ICLR.cc/2026/Conference — ICLR 2026 Conference Withdrawn Submission_

### Official Review · Reviewer_KP8x · 2025-10-30

**Soundness:** 2
**Presentation:** 1
**Contribution:** 2
**Rating:** 2
**Confidence:** 5

**Summary:**

This paper investigates the effect of integrating multiple models and multiple verifiers (reward models) under the Test-time Scaling framework, combined with self-repetition mechanisms such as Best-of-N (BoN). Building on these insights, the authors propose a method that ensembles multiple LLMs and multiple reward models. Compared with traditional parallel approaches like BoN, the proposed method demonstrates performance advantages in experiments.

**Strengths:**

1. The paper focuses on the timely and popular topic of TTS and extends it by considering multi-model ensembling. Empirical pre-test results show that incorporating multiple models and verifiers brings improvements in TTS performance, indicating practical significance.

2. The proposed empirical insights are exciting: under appropriate strategies, multi-model and multi-verifier settings outperform single-model or single-verifier baselines. These findings lay a foundation for further improving TTS systems.

**Weaknesses:**

1. The overall presentation and clarity require improvement.
+ (1) Key concepts are insufficiently defined and wrongly used. The paper restricts the definition of TTS to the “agent” setting, whereas most prior work considers LLMs more generally without limiting to agent-based contexts. Furthermore, it describes TTS as a two-stage process, parallel answer generation followed by selection, which only corresponds to parallel TTS and omits other paradigms such as tree search and self-refinement. The paper frequently narrows broad, well-established concepts to its own limited scope of interest, which undermines conceptual rigor and weakens the overall academic soundness.
+ (2) Problem formulation is missing, making it difficult for readers to understand the assumptions and applicability of the proposed method.
+ (3) Terminology is inconsistent. The term selector is used early but later replaced by the reward model without explanation. The more standard term verifier should be consistently adopted, as a reward model is only one possible implementation.
+ (4) Misinterpretation of collaboration. The manuscript frequently describes multiple agents independently producing outputs and then aggregating them as collaboration, but in my understanding, collaboration typically implies role differentiation and interaction, not mere independent execution followed by comparison.
+ (5) Ambiguity between “Agent” and “LLM.” The paper consistently uses Agent, yet the method does not highlight features specific to agents, such as role-playing or environment interaction. The term LLM would be more accurate.
+ (6) Paper writing structure requires revision. Section 4.2 presents preliminary experiments that motivate the method; such results should appear before the method section to ensure logical flow.
+ (7) The introduction lacks a high-level motivation and design overview of the proposed method.
+ (8) Inconsistencies and formatting issues: The introduction alternates between referring to “three” and “four” paradigms; Section 3.3.3 states that multiple reward models can be used, but the corresponding formula involves only one; inconsistencies appear in reference formatting (e.g., capitalization at line 595, missing authors at line 637).
+ (9) Several appendix sections are not cited in the main text.

2. Limited methodological novelty.
+ (1) The proposed approach is essentially a straightforward combination of LLM ensemble and RM ensemble, without addressing any new challenges like proposing an innovative mechanism arising from their integration.
+ (2) The motivation for model selection in the ACS component is unclear. Since the second stage (via RMs) already performs output selection, the first stage should aim to maximize diversity rather than identify the “best” model; explicit model selection may therefore be unnecessary.
+ (3) The RM selection strategy is naive, lacking technical depth or analysis of its underlying challenges.
+ (4) The reliance on a question pool for RM selection is unrealistic, as the source and availability of such a pool in practical scenarios are not specified.

3. Experimental design and baselines are insufficient.
+ (1) The paper does not specify the compute budget (e.g., number of parallel samples $n$), nor explain how fairness across methods under the same TTS budget is ensured.
+ (2) Baseline coverage is incomplete. The study lacks baselines for alternative ACS designs, diverse RM selection strategies, and other TTS paradigms such as tree search and self-refinement. The absence of such comparisons weakens the validity of method's effectiveness.
+ (3) The paper omits key experimental analyses, including sensitivity to the number of RMs ($k$), scaling behavior as test-time budget increases, and case studies.

**Questions:**

1. In the multi-model (MA) setup, is each LLM restricted to producing only one output? Has the trade-off between the number of models and the number of outputs per model been analyzed?
2. What are the experimental details for Figure 2? How is fairness ensured under equal computation budgets?
3. Can the proposed method be extended to self-refinement-style TTS? What modifications would be required?
4. Please provide a clear formal definition of the problem setting (inputs, outputs, available resources, budget constraints, and evaluation metrics).
5. How is the *MoR* term in Equation (1) designed or learned?
6. How is the parameter *k* chosen in Equation (2)? What tuning strategy is applied?
7. Does the reported inference time include model search and RM search/selection costs? Can the time distribution be broken down more precisely?
8. What is the source, size, and representativeness of the question pool?
9. Why does the paper only consider models with ≥20B parameters? Are smaller models compared or discussed?
10. The manuscript claims to provide an anonymous code repository, but no link is found. Please include the repository and full experimental configurations to ensure reproducibility.

---

### Official Review · Reviewer_B7qc · 2025-10-30

**Soundness:** 2
**Presentation:** 3
**Contribution:** 2
**Rating:** 2
**Confidence:** 4

**Summary:**

In this paper, the authors propose Collective Test-Time Scaling (CTTS), a framework that generalizes traditional test-time scaling (TTS) from the usual single-agent / single-reward-model paradigm to combining multiple agents and multiple reward models. The authors formalize three CTTS paradigms: SA–MR, MA–SR, MA–MR, and show empirically that using multiple agents and reward models (MA–MR) paradigm performs empirically best. Based on this observation, they propose CTTS-MM, which combines (1) Agent Collaboration Search (ACS), a greedy, reward-guided search procedure (with a residual aggregation fallback) to synthesize answers from multiple agent outputs — and (2) Mixture of Reward models (MoR), a retrieval-based reward-model selection based on Pairwise Reward Ranking (PPR) and Prior Reward model Ensemble Selection (PRES). To empirically validate the approach, the authors show experiments on a variety of benchmarks (mathematical reasoning, coding, knowledge QA, instruction following). The results indicate that CTTS-MM lead to consistent gains over baseline TTS techniques.

**Strengths:**

1. The paper in general is well-presented, and easy to follow. The research problem has been well-articulated to the reader.

2. The paper offers a comprehensive analysis of the performance across various configurations, including single-agent single-reward, single-agent multi-reward, and multi-agent multi-reward test-time scaling approaches.

3. The experimental evaluation is comprehensive, covering diverse benchmarks such as mathematical reasoning, coding, knowledge-based question answering, and instruction following tasks.

**Weaknesses:**

1. The primary concern lies in the limited technical novelty of the proposed method. Based on the established ensembling literature [1–5], it is intuitive to assume that leveraging multiple agents would naturally enhance performance. Moreover, even in the context of test-time scaling, recent studies have also validated this assumption [5,6].
In addition, the proposed agent-collaborative framework essentially performs a greedy search, and the reward model selection relies on dot-product similarity. To strengthen the technical contribution, the authors should include comparison with a simple baseline: such as generating n candidates from all agents and applying different aggregation functions (such as mean, median or min/max) to ensemble the rewards.

2. The comparison table also appears to be unfair. Although I appreciate the thorough analysis, comparing a multi-agent approach with single-agent baselines does not provide a balanced evaluation. I encourage the authors to instead compare their proposed approach against existing multi-agent baselines [5,6].


>References

1. Ensemble methods in machine learning
2. Simple and scalable predictive uncertainty estimation using deep ensembles
3. Neural ensemble search for uncertainty estimation and dataset shift
4. Helping or Herding?Reward Model Ensembles Mitigate but do not Eliminate Reward Hacking
5. Two Heads are Better Than One: Test-time Scaling of Multi-agent Collaborative Reasoning
6. Multi-Agent Verification: Scaling Test-Time Compute with Multiple Verifiers.

**Questions:**

Please refer weaknesses.

---

### Official Review · Reviewer_t7r8 · 2025-10-31

**Soundness:** 3
**Presentation:** 2
**Contribution:** 3
**Rating:** 4
**Confidence:** 2

**Summary:**

This paper proposes Collective Test-Time Scaling, a framework that generalizes single-model test-time scaling to multi-model and multi-reward collaboration. The authors introduce three paradigms and identify the multi-agent, multi-reward setting as the most effective. The resulting method, CTTS-MM, combines Agent Collaboration Search and Mixture of Reward Models to achieve significant improvements over Best-of-N, Self-Consistency, and even proprietary models such as GPT-4.1 across multiple benchmarks.

**Strengths:**

1. The paper clearly formalizes collective test-time scaling and provides a systematic comparison of different collaboration paradigms.

2. The proposed CTTS-MM integrates model search and reward aggregation in a coherent and well-motivated framework.

3. Experiments are extensive, covering seven benchmarks, ten models, and eight reward models, with consistent and strong results.

4. The ablation and efficiency analyses are detailed and help demonstrate the contribution of each component.

**Weaknesses:**

1. The paper lacks theoretical explanation of why multi-agent–multi-reward scaling leads to consistent improvements.

2. The robustness of the framework to inaccurate or biased reward models is not studied.

3. The ablations do not include simple baselines such as random selection or uniform reward weighting.

**Questions:**

see weakness.

---

### Official Review · Reviewer_oADX · 2025-10-31

**Soundness:** 3
**Presentation:** 2
**Contribution:** 2
**Rating:** 4
**Confidence:** 3

**Summary:**

This paper proposes Collective Test-Time Scaling (CTTS), a framework that extends beyond traditional Single Test-Time Scaling (STTS) by exploring three paradigms: SA-MR, MA-SR, and MA-MR. The authors identify MA-MR as the optimal paradigm and introduce CTTS-MM, which combines Agent Collaboration Search (ACS) for LLM ensemble selection and Mixture of Reward Models (MoR) with Prior Reward model Ensemble Selection (PRES) for adaptive reward model weighting. Experiments across seven benchmarks claim superiority over STTS methods and proprietary LLMs.

**Strengths:**

- **Originality**: First work to formalize and systematically explore CTTS paradigms.
- **Quality**: Rigorous experimentation across seven diverse benchmarks, with clear performance gains.
- **Clarity**: Well-organized methodology and ablation studies.
- **Significance**: Demonstrates that collective scaling can surpass flagship proprietary models using only open-source components, highlighting a promising research direction.

**Weaknesses:**

- **Experimental Breadth**: While seven benchmarks are used, they primarily focus on reasoning and coding tasks; inclusion of more diverse NLP tasks (e.g., dialogue, summarization) would strengthen generalizability claims.
- **Technical Details**: Some algorithmic details are vague (e.g., the exact implementation of the aggregator in ACS, hyperparameter sensitivity analysis).
- **Computational Cost**: Although inference time is discussed, a deeper analysis of the computational overhead of ACS and MoR is needed for practical applicability.

**Questions:**

1. The method appears highly dependent on the quality of the reward function. For more open-ended questions (e.g., tell me a joke), how can one obtain high-quality reward functions?
2. The aggregator Agg for summarizing answers from multiple models is crucial. How exactly is the aggregator Agg implemented?  And what attempts did the author make?
3. Did the authors attempt to use MoR in the post-training phase with reinforcement learning?
4. Section 3.3.3 is somewhat difficult to follow. Could the authors provide a more detailed explanation and clarify the motivation behind the specific design choices in the PRES algorithm? For example, each question has a cosine similarity vector; how is the "highest similarity score" derived from this vector? Is it a sum? Why is the selection based on similarity? What's the meaning of symbol $m$ in line 307-308? Does the algorithm iterate through all possible permutations and combinations of reward models in the pool? If so, wouldn't the computational overhead be prohibitively large?
5. Does the inference time reported in Table 4 include the time consumed by both the agent collaboration search (ACS) and the reward model search/selection (MoR/PRES) processes?
6. Have there been any experiments or attempts to apply this framework to multimodal tasks?
7. Can this method increase the upper limit of the model's capabilities? For example, if all individual agents answer incorrectly, can this method help them answer correctly?

---

### Note · Authors · 2026-01-12

I have read and agree with the venue's withdrawal policy on behalf of myself and my co-authors.